# Boson-peak-like anomaly caused by transverse phonon softening in strain glass

Shuai Ren [1,3], Hong-Xiang Zong [2,3], Xue-Fei Tao[2], Yong-Hao Sun [1], Bao-An Sun[1], De-Zhen Xue[2], Xiang-Dong Ding [2✉] & Wei-Hua Wang [1✉]

Strain glass is a glassy state with frozen ferroelastic/martensitic nanodomains in shape memory alloys, yet its nature remains unclear. Here, we report a glassy feature in strain glass that was thought to be only present in structural glasses. An abnormal hump is observed in strain glass around 10 K upon normalizing the specific heat by cubed temperature, similar to the boson peak in metallic glass. The simulation studies show that this boson-peak-like anomaly is caused by the phonon softening of the non-transforming matrix surrounding martensitic domains, which occurs in a transverse acoustic branch not associated with the martensitic transformation displacements. Therefore, this anomaly neither is a relic of van Hove singularity nor can be explained by other theories relying on structural disorder, while it verifies a recent theoretical model without any assumptions of disorder. This work might provide fresh insights in understanding the nature of glassy states and associated vibrational properties.

[1] Institute of Physics, Chinese Academy of Sciences, 100190 Beijing, China. [2] State Key Laboratory for Mechanical Behavior of Materials, Xi'an Jiaotong University, 710049 Xi'an, China. [3] These authors contributed equally: Shuai Ren, Hong-Xiang Zong. ✉email: dingxd@mail.xjtu.edu.cn; whw@iphy.ac.cn

Phase transitions in ferroic materials involve long-range ordering of a certain order parameter. However, when the frustration caused by point defects or dopants is strong enough in the system, a metastable glassy state with local order of the order parameter may take place[1]. The order parameter can be magnetic moment (for ferromagnetic transition), dielectric polarization (for ferroelectric transition) and lattice strain (for ferroelastic/martensitic transition), which corresponds to a different ferroic glassy state, i.e., spin glass, ferroelectric relaxor and strain glass, respectively[1–4].

In comparison with spin glass and ferroelectric relaxor, strain glass is a relatively new state of matter in concept, which is a glassy state with local strain order while maintaining average structure unchanged in shape memory alloys[5]. Strain glass is of interest due to its unique functionalities[5–9] such as slim or even non-hysteretic superelasticity and low-field-triggered large magnetostriction, and thereby is of potential importance to develop novel smart materials. The glass behavior of strain glass has been characterized by the slowing-down of dynamics[4,10,11], and some phenomenological models have been established to successfully capture the experimental features of strain glass[12,13]. However, despite these efforts, the nature of strain glass is still unclear; especially the glassy dynamic features of strain glass are still blurred, as compared with those of metallic glass, the well-known structural glass in metals.

Due to lacking crystalline periodicity, metallic glass exhibits unique dynamic relaxations distinct from the corresponding crystalline alloys, such as primary α-relaxation, β-relaxation, and boson peak[14–16]. In particular, the boson peak is a universal glassy dynamic feature found in metallic glass (and other structural glasses)[14–16], which is an anomaly in the vibrational density of states (VDOS), where an excess of vibrational states takes place, departing from the Debye squared-frequency law for crystals at low frequencies of the order of 1 THz. This anomaly becomes an anomalous peak upon plotting the reduced VDOS $g(\omega)/\omega^2$ over $\omega$ ($\omega$ is the frequency), and can also be manifested by a peak in the specific heat ($C_p$) at 5–20 K in a plot of $C_p/T^3$ vs. $T$.

It is widely accepted that the boson peak reflects the intrinsic vibrational properties of structural glasses, and thus plays a key role in fundamentally understanding the nature of structural glasses. After wide investigations for decades, various models have been proposed to interpret the origin of the boson peak, such as the heterogeneous elasticity theory[17], soft anharmonic potentials[18,19], phonon-saddle transition in the energy landscape[20], local inversion-symmetry breaking associated with nonaffine shear softening[21,22], and smeared out van Hove singularity[23,24]. As a result, the origin of the boson peak is still under debate. On the other hand, there are emerging reports recently exhibiting a similar boson peak anomaly in some crystalline systems with less or even no disorder, such as atomic/molecular cryocrystals[25], halomethanes[26], organic crystals[27], and quasicrystals/incommensurate systems[28]. It calls for a deeper understanding of the origin of the boson peak since all the models in structural glasses proposed so far rely on assumptions of some form of disorder and cannot interpret the boson peak in these crystalline systems. Intriguingly, a recent work proposed a promising theoretical framework without any assumptions whatsoever of disorder, which provides the first theoretical explanation for the boson peak in these systems with less or no disorder[29,30]. The emergence of the boson peak in some crystalline systems indicates that the boson peak is a universal glassy anomaly beyond the range of structural glasses, which makes the origin of the boson peak even more complicated.

The strain glass is promising to provide a fresh perspective for understanding the origin of the boson peak. From the view of microstructure, strain glass is fundamentally distinct from metallic glass. For metallic glass there is an amorphous structure, whereas strain glass is a glassy phenomenon in crystalline alloys (see Supplementary Fig. 1). For example, the average structure of strain glass in Ti–Ni alloys is a simple B2 structure[5]. Despite the fundamental difference in atomic configuration, they may share similar glassy dynamic features; for example, both of them undergo a slowing-down of dynamics during the glass transition. In this study we try to link these two glasses in alloys from the view of phonon dynamics by exploring whether strain glass possesses a similar boson peak behavior, which can not only solidify strain glass as a glassy state of matter but help to deepen our generic understanding of glassy phenomena as well.

Here we found an abnormal hump around 10 K in the plot of $C_p/T^3$ vs. $T$ in strain glass, which resembles the boson peak of metallic glass. Molecular dynamic (MD) simulation revealed that the boson-peak-like (BP-like) anomaly corresponds to excess vibrational modes at low frequencies in its VDOS, and the excess modes stem from the phonon softening of the non-transforming regions surrounding the martensitic nanodomains, which takes place in the transverse acoustic (TA) branch not associated with the martensitic transformation displacements. We further found the BP-like anomaly gradually becomes weak with the weakening of phonon softening by simulation, which nicely aligns with the prediction of the recent theoretical model with no assumption of disorder. This work found a BP-like anomaly caused by TA phonon softening in strain glass, which may not only shed light on the nature of strain glass but help to deepen the understanding of the boson peak in both disordered and ordered solids as well.

## Results

**The BP-like anomaly in strain glass.** $Ti_{50-x}Ni_{50+x}$ alloys (abbreviated by $x$Ni hereafter) are chosen in the main text, because they serve as the prototypical system of strain glass[1]. It has been reported that strain glass appears at $x \geq 1.5$ in $Ti_{50-x}Ni_{50+x}$ alloys[31], and the phase diagram established in this work (see Supplementary Fig. 2) is consistent with the one in the previous work[31].

Figure 1a shows the temperature dependence of $C_p$ for different $x$. All the curves are asymptotically close to the expected classical value of Dulong-Petit law ($3R$ per mole of atoms in metals, where $R$ is the ideal gas constant). There are clear peaks in $C_p$ for 0.6Ni and 1Ni, respectively. As for 0Ni, the martensitic transformation temperature is above 300 K, so there is no peak observed in the black curve. When the alloys enter into the strain glass range ($x \geq 1.5$), the peak in $C_p$ disappears, while a hump around 200 K is detected instead, when taking the black curve of 0Ni as a baseline. Thus, the $C_p$ measured by the thermal relaxation method shows that the strain glass transition is still accompanied by a broad and sluggish caloric change, and such a change gradually becomes weak as $x$ increases.

To show how $x$ influences the $C_p$ at low temperatures, the $C_p$ vs. $T$ is plotted in the logarithmic scale in Fig. 1b. The seven curves are classified into two groups, i.e., the martensitic phase ($x < 1.5$) and strain glass ($x \geq 1.5$). There is a jump observed in $C_p$ between the martensitic phase and strain glass at low temperatures. Interestingly, a similar jump in $C_p$ can also be found between a $Zr_{50}Cu_{40}Al_{10}$ bulk metallic glass and the corresponding crystalline alloy at low temperatures, as shown in the inset of Fig. 1b. It indicates that strain glass and metallic glass share a similar $C_p$ anomaly at low temperatures.

For metals, the specific heat is mainly contributed by electrons and phonons at low temperatures. As shown in Fig. 1c, the specific heat below 7 K can be fitted in a plot of $C_p/T$ vs. $T^2$ by the relation $C_p/T = \gamma + \beta T^2$, where the intercept $\gamma$ is the electronic specific heat coefficient, and the slope $\beta$ is the calorimetric cubic

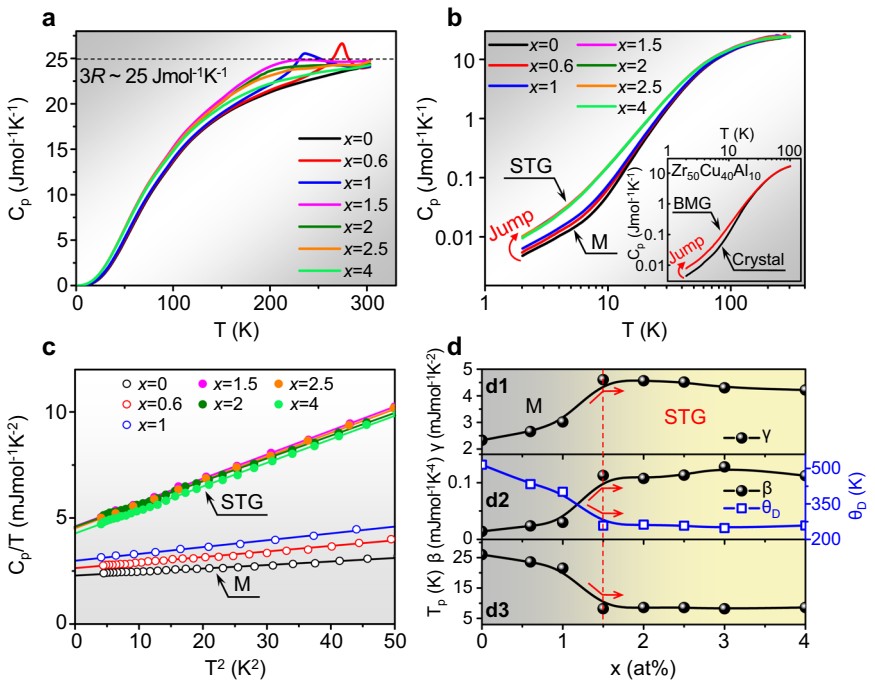

**Fig. 1 Specific heat ($C_p$) and thermal parameters in strain glass alloys. a** The $C_p$ curve for each $x$ in $Ti_{50-x}Ni_{50+x}$. The dashed line represents the classical value of Dulong-Petit law. **b** $C_p$ as a function of temperature in the logarithmic scale. A jump in $C_p$ can be observed from the martensite (M) to strain glass (STG) at low temperatures. The inset exhibits a similar jump between $Zr_{50}Cu_{40}Al_{10}$ bulk metallic glass (BMG) and its corresponding crystal. **c** $C_p/T$ vs. $T^2$ relation for each $x$ with the $C_p$ data below 7 K. The linear fitting follows the formula of $C_p/T = \gamma + \beta T^2$. **d** The composition dependence of thermal parameters, including (**d1**) the electronic specific heat coefficient $\gamma$, (**d2**) the calorimetric cubic coefficient $\beta$ and the Debye temperature $\Theta_D$, and (**d3**) the peak temperature ($T_p$) in the plot of reduced specific heat ($C_p-\gamma T)/T^3$ vs. $T$ in Fig. 2. The dashed line and the red arrows illustrate the crossover between the martensite and strain glass.

coefficient. Then, we can derive the change of thermal parameters with $x$ from Fig. 1c.

The composition dependence of the thermal parameters is summarized in Fig. 1d. As indicated by the red arrows, both $\gamma$ and $\beta$ exhibit a sharp increase at the crossover between the martensite and strain glass ($x = 1.5$). On the other hand, the Debye temperature $\Theta_D$ can be derived from $\beta$ through the relation $\beta = 12\pi^4 R/(5\Theta_D^3)$, so $\Theta_D$ shows a sharp decrease at the crossover. Therefore, the strain glass possesses a higher $\gamma$ and $\beta$, and a lower $\Theta_D$ than the martensite, which is the same as the property change tendency between the metallic glass and its corresponding crystal[32].

To exhibit the $C_p$ anomaly caused by the lattice dynamics more clearly, the ($C_p-\gamma T)/T^3$ vs. $T$ is plotted in the logarithmic scale in Fig. 2. A pronounced peak can be found around 30 K in the martensitic phase for each $x$, which has been well explained by the lattice dispersions such as van Hove singularities[33,34]. In comparison, strain glass exhibits a broad hump around 10 K, and the maximum is much higher than the peak value of the martensitic phase. Thus, a large gap is observed between the martensitic phase and strain glass below 30 K. The peak temperature ($T_p$) as a function of $x$ is shown in Fig. 1d3. A sharp decrease is observed at the crossover, and the $T_p$ of strain glass is almost a constant around 10 K. The hump of strain glass around 10 K is quite similar to the boson peak of the $Zr_{50}Cu_{40}Al_{10}$ bulk metallic glass in Supplementary Fig. 3 as well as other metallic glasses[35]. Moreover, it is noted that this BP-like hump is also found in other strain glass alloys like $Ti_{50}Pd_{50-x}Cr_x$ (see Supplementary Fig. 4), which means that the BP-like hump is a universal anomaly in strain glass alloys.

**The BP-like anomaly reproduced by simulation.** In the conventional understanding of the nature of strain glass, when the

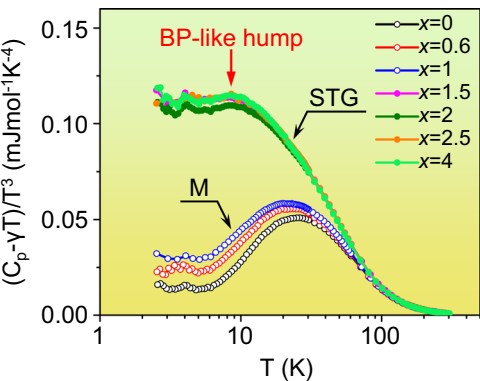

**Fig. 2 BP-like anomaly of strain glass in the reduced specific heat ($C_p-\gamma T)/T^3$ as a function of $T$.** The hump of strain glasses around 10 K is similar to the boson peak of metallic glass, and thus termed as the BP-like hump.

system undergoes the slowing-down of dynamics during the strain glass transition upon cooling, the martensitic nanodomains are already frozen, and no other glassy dynamic feature is expected to appear below the glass transition temperature[1]. Thus, it is an interesting mystery why the BP-like anomaly can appear at low temperatures in strain glass.

We first try to reproduce the BP-like anomaly in strain glass. An MD simulation is performed to calculate the VDOS and $C_p$ of a model alloy system, which has been confirmed as a reliable model system for strain-glass-involved simulation in the previous work[36]. Figure 3a shows the calculated VDOS as a function of frequency for different defect concentrations in this model system ($Zr_{100-x}Ni_x$). According to the previous work, the critical defect

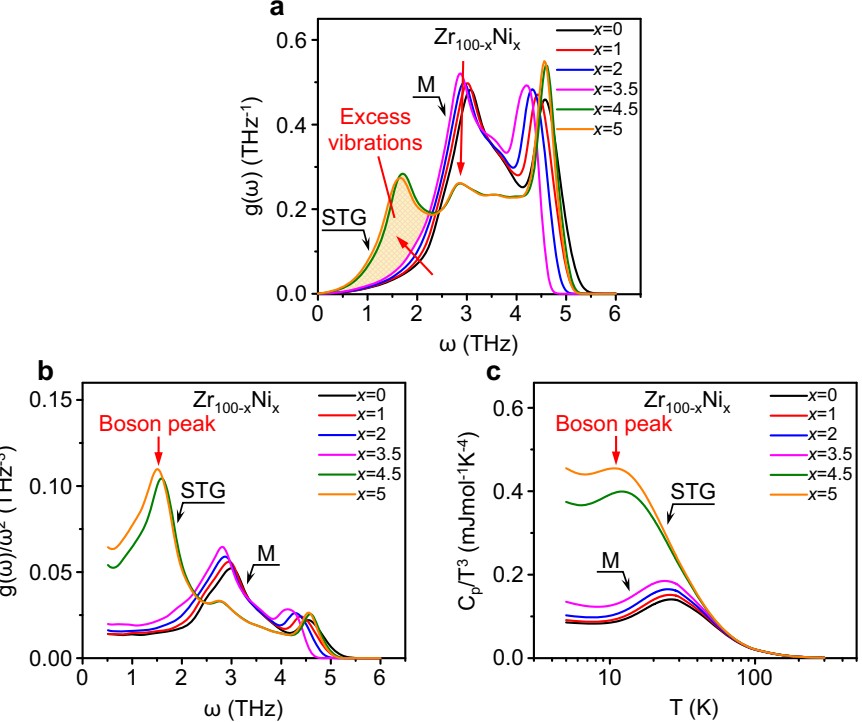

**Fig. 3 BP-like anomaly reproduced by simulation. a** Vibrational density of states (VDOS) $g(\omega)$ as a function of frequency $\omega$ for each $x$ in a strain glass model system ($Zr_{100-x}Ni_x$). **b** Reduced VDOS $g(\omega)/\omega^2$ as a function of $\omega$ for each $x$. The boson peak is observed around 1.5 THz for strain glass. **c** The plot of $C_p/T^3$ vs. $T$ calculated by the VDOS.

concentration for strain glass is around $x = 4$ (see Supplementary Fig. 5)[36]. Accordingly, the VDOS curves in Fig. 3a are divided into two groups, i.e., the martensite ($x < 4$) and strain glass ($x > 4$). The martensitic phases exhibit similar VDOS curves with the first peak around 3 THz. This peak is known as the TA van Hove singularity in crystals[33,34]. In comparison, the first peak of strain glass appears at a much lower frequency (~1.5 THz), at the expense of a strong decrease of the second peak which corresponds to the TA van Hove singularity of the martensitic phase. Although the peak intensity around 3 THz strongly decreases from the martensite to strain glass, the shift of the peak frequency is negligible. It means that the TA van Hove singularity of the martensitic phase persists in strain glass. Therefore, the possibility that the first peak of strain glass is generated by the shift of TA van Hove singularity of the corresponding martensitic phase can be ruled out.

The reduced VDOS $g(\omega)/\omega^2$ as a function of $\omega$ for each $Zr_{100-x}Ni_x$ model alloy is shown in Fig. 3b. For perfect crystals, the low-frequency VDOS obeys the Debye squared-frequency law, i.e., $g(\omega)$ is proportional to $\omega^2$. Thus, the reduced VDOS $g(\omega)/\omega^2$ of a perfect crystal is a constant at low frequencies. The reduced VDOS of the martensite in Fig. 3b keeps constant below 2 THz, consistent with the Debye law. It then deviates from the Debye model, as the van Hove singularity (around 3 THz) is approached. In comparison, a peak is observed around 1.5 THz in the reduced VDOS of strain glass, which is much lower than the frequency of the van Hove singularity of the martensite, and locates in the similar frequency range of the boson peak in metallic glass. Therefore, we also termed this peak as the boson peak in strain glass.

Figure 3c shows the $C_p$ calculated by the VDOS of each alloy in the plot of $C_p/T^3$ vs. $T$. It is noted that the calculated curves well reproduce the experimental features of both martensitic phase and strain glass in Fig. 2, and especially the BP-like hump of

strain glass is confirmed by simulation. As a result, Fig. 3 indicates that the BP-like anomaly in the $C_p/T^3$ of strain glass is inherently tied to the excess low-frequency vibrations in its VDOS.

**The structural origin of the BP-like anomaly**. To further figure out the structural origin of the boson peak in strain glass, we investigated the correlation of the additional low-frequency modes below 2 THz in the VDOS with the microstructure of strain glass through the MD simulation. The prevailing microstructural picture of strain glass is described as isolated martensitic nanodomains distributed randomly in the parent phase matrix[1,12,13]. Recently, the microstructural picture has been further developed to be that the martensitic nanodomains are surrounded by percolated strain networks in the matrix[36]. In either way, it is important and intriguing to explore which part in the microstructure gives rise to the excess vibrational modes in the VDOS of strain glass.

We employed atomic pinning methods to extract each VDOS of the martensitic nanodomains and their surrounding nontransforming matrix in a strain glass model alloy (Zr–4.5%Ni). As shown in Fig. 4a, the nanodomains show a VDOS feature analogous to that of the martensitic phase in Fig. 3a. In contrast, the non-transforming matrix exhibits a similar VDOS feature to the strain glass in Fig. 3a, in which excess low-frequency modes are observed. It indicates the important role of the nontransforming matrix in the presence of BP-like anomaly. Figure 4b exhibits the calculated phonon dispersion curves of this alloy. Obvious phonon softening is observed in the [001] TA branch (the red arrow in Fig. 4b), and the frequency of the boson peak ($\omega_{BP}$) corresponds to the frequency of the phonon softening (~1.6 THz). Therefore, Figures 4a, b together indicate that the excess low-frequency vibrations corresponding to the BP-like

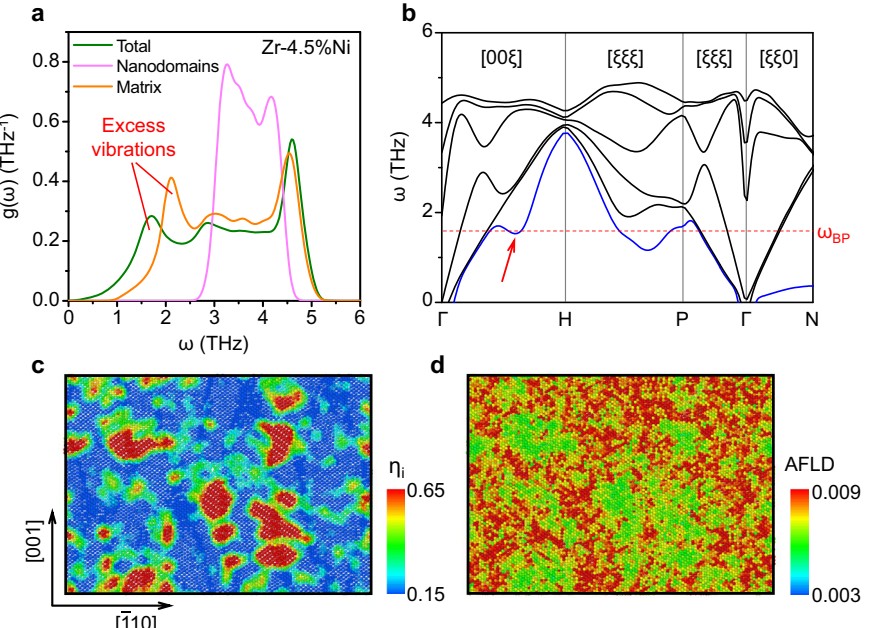

**Fig. 4 Structural origin of the BP-like anomaly in strain glass. a** VDOS of nanodomains, parent phase matrix as well as the entirety for a strain glass model alloy (Zr–4.5%Ni). **b** Phonon spectra of this strain glass alloy at 40 K. The red arrow marks the phonon softening along [001], and the red dash line denotes $\omega_{BP}$. **c** Spatial distribution of order parameter ($\eta_i$) at the (110) plane for this alloy at 40 K. Red regions correspond to martensitic nanodomains. **d** Spatial distribution of auto-correlation function of local atomic displacement (AFLD) at 40 K. High AFLD regions (colored red) indicate a high density of excess low-frequency phonon modes.

anomaly are attributed to the phonon softening of the [001] TA branch, which occurs in the non-transforming matrix.

Figures 4c, d further exhibit the spatial correlation between the local structure (Fig. 4c) and the corresponding density of the excess low-frequency phonon modes (Fig. 4d) in this strain glass. Here, the local structure or phase is characterized by the order parameter $\eta_i$, while the local density of the excess low-frequency phonon modes is estimated by the auto-correlation function of local atomic displacement (AFLD), i.e., $<d(0)*d(\tau)>$, where $d$ is the local atomic displacement, and $1/\tau$ is the characteristic frequency of the excess low-frequency phonon modes ($1/\tau = 1.63$ THz). In Figs. 4c, d, the spatial correlation between the two quantities is apparent: the region with a lower order parameter (colored blue in Fig. 4c), corresponding to the parent phase matrix, has a higher level of AFLD (colored red in Fig. 4d). It means that the parent phase matrix has a high density of excess low-frequency phonon modes. Therefore, the microstructure of strain glass can be regarded as "hard" nanodomains surrounded by the "soft" non-transforming matrix from the aspect of the vibrational degree of freedom. This result further supports that the BP-like anomaly in strain glass stems from the "soft" non-transforming matrix, rather than the martensitic nanodomains.

## Discussion

Although strain glass shares a similar boson peak anomaly with structural glasses, the mechanism underlying the BP-like anomaly of strain glass seems different. Over time, several competing models[17–24] have been proposed to interpret the origin of the boson peak in structural glasses, which can be roughly classified into two frameworks. One is considered that the excess modes are (quasi)localized and arise from the phonon damping related to structural disorder[17–22]. The other was recently proposed that the boson peak modes of the glass come from the broadening and shift of the TA van Hove singularity of the corresponding crystalline state[23,24]. Consequently, a unified and coherent conclusion has not been reached in structural glasses.

We find that both of the frameworks in structural glasses fail to explain the BP-like anomaly in strain glass. It is easy to figure out that the model of the shift of TA van Hove singularity is not applicable to strain glass. As shown in Fig. 3a, the excess modes giving rise to the BP-like anomaly in strain glass exhibit a clear separation from the TA van Hove singularity of the corresponding martensitic phase. Consequently, the excess vibrations are not from the shift of TA van Hove singularity. It means that the boson peak in strain glass is not a relic of the van Hove singularity, which is consistent with the conclusion in structural glasses reported by other authors recently[22].

On the other hand, in the models based on localized vibrational modes in structural glasses, the appearance of the localized modes is generally considered to be attributed to strong phonon damping due to structural disorder. The phonon damping can decrease the mean free path of phonons. When the phonon damping is strong enough, the mean free path of phonons becomes smaller than their wavelength, then these strongly damped lattice waves cease to exist as well-defined phonons. The crossover is known as the Ioffe–Regel limit, at which the mean free path is comparable to the wavelength. It can also be treated equivalently in terms of frequency: above the Ioffe–Regel crossover frequency $\omega_{IR}$, the phonons become ill-defined phonons or damped phonons. It has been demonstrated that $\omega_{BP}$ is equivalent to the Ioffe–Regel limit for the transverse acoustic wave ($\omega_{IR-TA}$) in structural glasses[37]. As a result, the boson peak frequency is believed to mark the characteristic frequency of transverse vibrational modes, where the transverse phonons transform from propagating to diffusive modes[37]. Interestingly, a recent neutron scattering work has found glasslike phonon damping in strain glass[38]. The phonon damping occurs in the mode with displacements consistent with the martensitic transformation (known as the $TA_2$ mode), whereas no TA broadening is found in the directions not associated with the transition displacements[38]. As a consequence, it is of importance to distinguish the different roles of the phonon softening and the phonon damping in strain

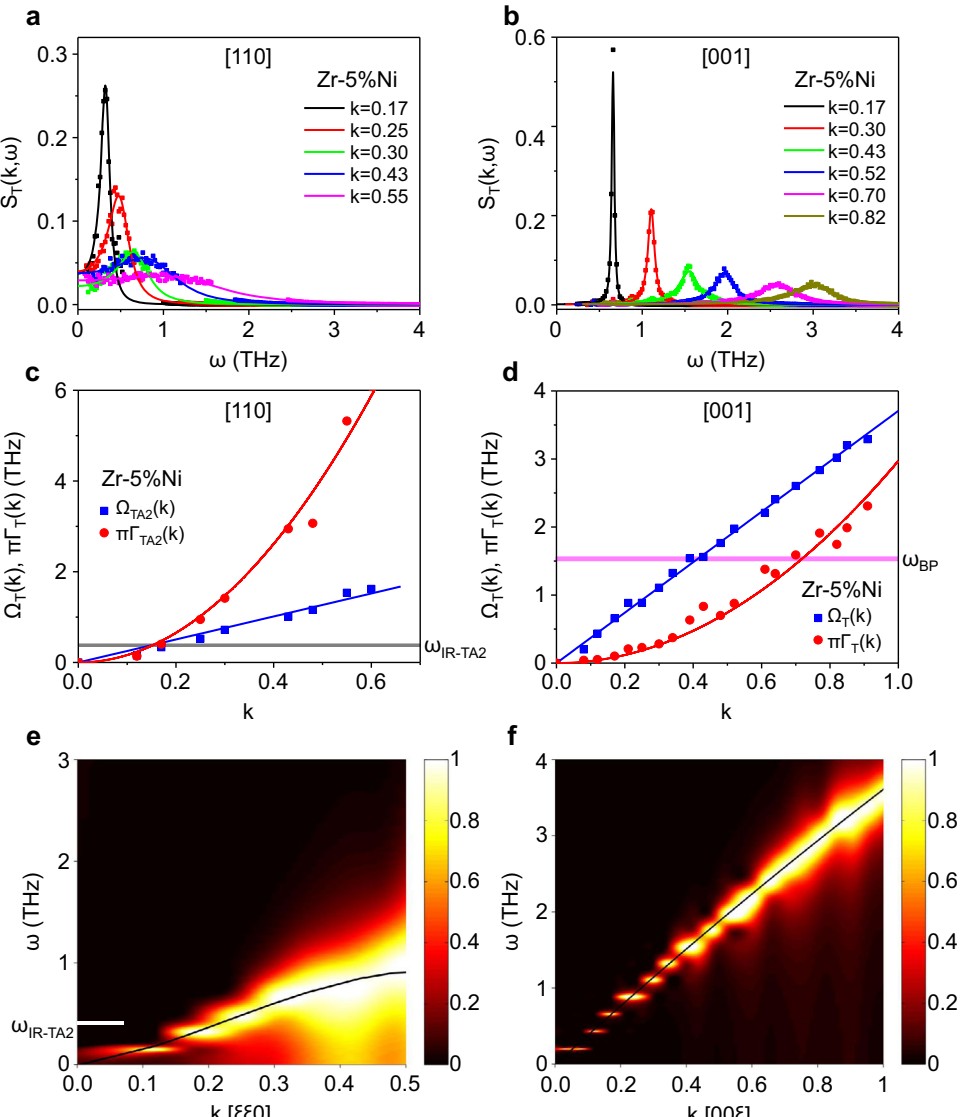

**Fig. 5 Transverse phonon dynamics and the dispersion relations. a, b** Transverse dynamical structure factors of the strain glass Zr–5%Ni along (**a**) [110] and (**b**) [001], respectively. Solid lines are the fits by the damped harmonic oscillator model. **c, d** Dispersion relation $\Omega_T(k)$ and excitation broadening $\pi\Gamma_T(k)$ for transverse motion along (**c**) [110] and (**d**) [001], respectively. **e, f** Contour maps of the normalized dynamical structure factor on the plane ($\omega, k$) for directions (**e**) [110] and (**f**) [001], respectively. The black full line corresponds to the dispersion relation fitted by the damped harmonic oscillator model. The white horizontal dash in (**e**) shows the transverse Ioffe–Regel crossover frequency $\omega_{\text{IR-TA2}}$ along [110].

glass, and the crucial piece of evidence is to discriminate the relation between $\omega_{\text{BP}}$ and $\omega_{\text{IR-TA}}$ in strain glass.

We thus calculate the $\omega_{\text{IR-TA}}$ of different directions by adopting a common calculation method[37]. Figures 5a, b exhibit the transverse dynamical structure factors $S_T(k, \omega)$ of strain glass (Zr–5%Ni) along [110] and [001] at 40 K, respectively (for more details, please refer to Supplementary Fig. 6). The solid lines in Figs. 5a, b are the fits by the damped harmonic oscillator model[37], which is

$$S_T(k, \omega) = \frac{A_T(k)}{\left[\omega^2 - \Omega_T^2(k)\right]^2 + \omega^2\Gamma_T^2(k)}, \quad (1)$$

where $\Omega_T(k)$ corresponds to the excitation frequency, $\Gamma_T(k)$ corresponds to the full-width at half-maximum of the excitations or the phonon damping coefficient, and $A_T(k)$ is the fitting coefficient. The obtained $\Omega_T(k)$ and $\Gamma_T(k)$ of different directions are plotted in Figs. 5c, d respectively. In general, $\Omega_T(k)$ obeys a linear dispersion relation with $k$, and $\Gamma_T(k)$ follows a $k^2$ law, which is consistent with

the literature (refs. [29,37] and references therein). The Ioffe–Regel limit condition is given by $\Omega_T(k) = \pi\Gamma_T(k)$, which corresponds to the intersection point of two curves in Figs. 5c, d.

It is noted that a very low $\omega_{\text{IR-TA2}}$ (~0.4 THz) is observed along [110] in Fig. 5c, which corresponds to the basal mode with displacements matching the martensitic transformation (i.e., the $TA_2$ mode). This result indicates strong phonon damping occurs in the $TA_2$ mode, which is consistent with the experimental results in the previous work[38]. In comparison, there is no intersection point in the first Brillouin zone along [001] in Fig. 5d, which corresponds to the soft mode but not associated with the transition displacements. This result also nicely aligns with the previous experimental results[38]. The absence of the intersection point means that the lattice waves are still well-defined phonons in the first Brillouin zone along [001], and thus $\omega_{\text{BP}}$ is unrelated to $\omega_{\text{IR-TA}}$ along [001].

For a better visual effect, we exhibit the contour maps of the normalized factor $S_T(k, \omega)$ on the plane ($\omega, k$) for the two

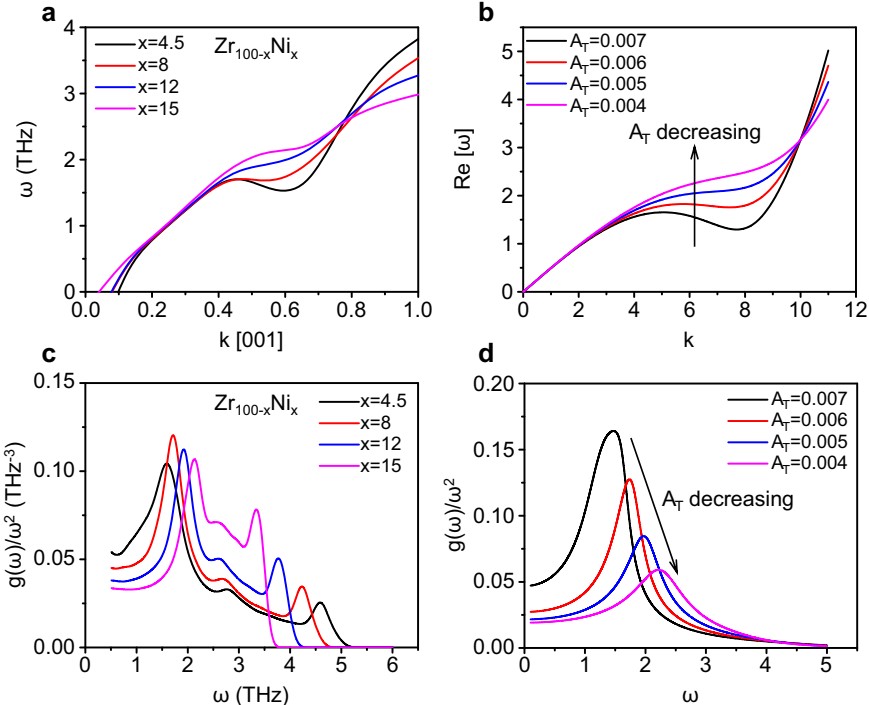

**Fig. 6 Comparison of phonon dynamics obtained by simulation and the theoretical framework without assumptions of disorder.** **a** [001] TA branches for different strain glass compositions ($x$) by simulation. **b** TA branches calculated by the Eq. (5). We set $\nu_T = 0.5$, $D_T = 0.012$ and dial $A_T$ in the range of [0.004, 0.007]. $B_T$ changes with $A_T$ in the range of [0.000055, 0.000025]. **c** Reduced VDOS $g(\omega)/\omega^2$ vs. $\omega$ for different $x$ by simulation. **d** Reduced VDOS $g(\omega)/\omega^2$ vs. $\omega$ according to the theoretical framework [Formula (3)]. We set $k_D = 11$. The contribution of longitudinal acoustic branch is omitted here.

directions, [110] and [001], in Figs. 5e, f. The normalized factor is obtained by dividing by the maximum of curves at each fixed $k$. It is obvious that the phonon broadening of the [110] branch is severe in Fig. 5e, while that of the [001] branch is very weak in Fig. 5f. Therefore, these simulation results along [110] and [001] clearly show that the BP-like anomaly is not associated with phonon damping. As a result, these results further lend credence to the mechanism that the BP-like anomaly in strain glass is caused by phonon softening.

This work indicates that it requires a complete re-thinking process about the origin of the boson peak in ordered solids. The structural disorder is generally believed to play an important role in understanding the origin of the boson peak in structural glasses[17–22], whereas this work suggests that the boson peak in ordered crystalline solids is not necessarily related to the structural disorder. Thus, those theoretical models relying on the assumptions of disorder in structural glasses fail in capturing the origin of the boson peak in these ordered solids. On the other hand, it is noted that an impressive theoretical model has been proposed recently without any assumptions of disorder, which represents the first theoretical framework to successfully explain the boson peak in ordered crystalline systems[29,30]. More interestingly, this theoretical framework is also found to nicely match our results.

According to this theoretical model, the vibrational properties of an isotropic solid can be generically represented by a Green's function[29,30],

$$G_\lambda(k, \omega) = \frac{1}{\omega^2 - \Omega_\lambda^{*2}(k) + i\omega\Gamma_\lambda^*(k)}, \qquad (2)$$

where $\lambda$ represents either the longitudinal $\lambda = L$ or transverse $\lambda = T$ displacement field. $\Omega_\lambda^*(k)$ is the eigenfrequency, corresponding to the energy of the acoustic phonons. $\Gamma_\lambda^*(k)$ is the

damping coefficient, representing the diffusive phonon damping for acoustic phonons, and goes quadratically with the wave vector, $\Gamma_\lambda^*(k) \sim k^2$.

Then, the VDOS of the system can be calculated by substituting the Green's function of Eq. (2) into the formula[30],

$$g(\omega) = -\frac{2\omega}{\pi k_D^3} \int_0^{k_D} \mathrm{Im}[2G_T(\omega, k) + G_L(\omega, k)]k^2 \mathrm{d}k, \qquad (3)$$

where $k_D$ denotes the maximum (Debye) wavenumber in the system. Since the boson peak is totally determined by the TA modes, for simplicity, we then neglect the longitudinal acoustic modes and focus on the TA branch. The TA dispersion relations of the system can then be written as[30],

$$\Omega_T^{*2}(k) = \nu_T^2 k^2 - A_T k^4, \qquad (4)$$

where $\nu_T$ is the phonon speed of propagation of the TA branch, related to the shear elastic moduli and $A_T$ is an effective parameter of the TA branch, representing anharmonic effects in the system.

On the other hand, the simulation results in this work exhibit that phonon softening takes place somewhere around the center of the Brillouin zone along [001], as shown in Fig. 6a. In order to capture the characteristic of phonon softening in simulation, we need to add an additional higher order term in the TA dispersion relations:

$$\Omega_T^{*2}(k) = \nu_T^2 k^2 - A_T k^4 + B_T k^6, \qquad (5)$$

where $B_T$ is a fitting parameter of the additional term. The TA dispersion relations obtained by Eq. (5) with decreasing the anharmonic parameter $A_T$ are shown in Fig. 6b. It is found that the TA dispersion curves in Fig. 6b are analogous to the simulation results in Fig. 6a. It means that by adding the additional term, Eq. (5) can capture the main characteristic of the TA

branches in simulation, except for the flattening of the branches upon approaching the Brillouin zone boundary in simulation.

We then calculate the reduced VDOS according to the dispersion relations from the simulation and the theoretical model, respectively. Figure 6c exhibits the reduced VDOS obtained by simulation. With the increase of Ni concentration $x$, the phonon softening along [001] in Fig. 6a becomes weak gradually. Accompanying this process, the boson peak in Fig. 6c shifts to a higher frequency. As a consequence, it is explicit that the boson peak in strain glass is determined by phonon softening. Intriguingly, the reduced VDOS calculated by the theoretical model in Fig. 6d is in accordance with the simulation results in Fig. 6c. The decrease of the anharmonic parameter $A_T$ causes the phonon softening to gradually become weak in Fig. 6b, which also leads to the increase of the boson peak frequency, as shown in Fig. 6d. As a result, the theoretical model exhibits a nice marriage with the simulation results. It means that this theoretical framework without any assumptions of disorder also well supports the mechanism of phonon softening as the origin of the BP-like anomaly in strain glass.

We note that the interpretation of the BP-like anomaly in strain glass does not necessarily apply to all glasses in general. The boson peak in structural glasses is defined as an excess of vibrational states in the VDOS as compared with either the Debye squared-frequency law or their long-range ordered counterpart (i.e., crystals)[20]. Similarly, the BP-like anomaly of strain glass in this work is defined as the excess states beyond its corresponding long-range ordered phase (i.e., martensitic phase). However, the underlying mechanisms of the two anomalies are different. For structural glasses, the boson peak is tightly related to phonon damping and takes place in a frequency region where the dispersion relation is still linear. In contrast, the BP-like anomaly in strain glass is tied to a nonlinear TA branch exhibiting phonon softening. The key differences can be presented in a more explicit way according to the Green's function of Eq. (2): the boson peak in structural glasses is mainly governed by a strong damping coefficient $\Gamma^*_T(k)$[29,30], whereas the BP-like anomaly in strain glass is mostly dominated by a nonlinear propagating term $\Omega^*_T(k)$. Our work does reveal that the BP-like anomaly can be triggered by a different mechanism but does not deny the fundamental role of phonon damping in understanding the origin of the boson peak in structural glasses.

In conclusion, we found a BP-like anomaly in Ti–Ni strain glass alloys, which is caused by phonon softening. A hump is observed around 10 K in the plot of $(C_p - \gamma T)/T^3$ vs. $T$, similar to the boson peak in metallic glass. The MD simulation reveals that this BP-like anomaly in $C_p$ corresponds to the excess vibrations at low frequencies, and the excess vibrations stem from the TA phonon softening of the non-transforming matrix surrounding martensitic nanodomains, rather than the nanodomains themselves. As a result, the boson peak is a common glassy feature in both strain glass and metallic glass, but it stems from a different mechanism in strain glass. The BP-like anomaly in strain glass is neither the relic of the van Hove singularity nor related to the phonon damping. Remarkably, our results exhibit a nice match with a recent theoretical framework without any assumptions of disorder. This work may provide fresh insight into the nature of strain glass, and more importantly, may lead to the opening of a fresh conception of glasslike anomalies in absence of structural disorder.

## Methods

**Sample preparation**. Samples of $Ti_{50-x}Ni_{50+x}$ ($x = 0$–4) and $Ti_{50}Pd_{50-x}Cr_x$ ($x = 0$–14) alloys were prepared from high-purity metals (>99.95 at%) by arc-melting under an argon atmosphere. The ingots were then cast into a copper-mold with a square size of $3 \times 3$ mm$^2$ in an argon atmosphere. The specimens were

solution-treated at 1273 K for 12 h in evacuated quartz tubes and then water quenched. Cylindrical rods of $Zr_{50}Cu_{40}Al_{10}$ bulk metallic glass (BMG, 2 mm in diameter) were fabricated by the same method of copper-mold casting in an argon atmosphere. The corresponding crystal of $Zr_{50}Cu_{40}Al_{10}$ was obtained through annealing the BMG at 823 K for 30 min. The specimens were then cut into desired sizes for different experiments.

**Property characterization**. The latent heat of transformation was measured by a differential scanning calorimeter (TA DSC-Q200 and Perkin-Elmer DSC-7) with a cooling/heating rate of 10 K/min. The possible strain glass transition was detected by a dynamic mechanical analyzer (TA DMA-Q800) using a step-cooling method at the single cantilever mode in the frequency range from 0.2 to 20 Hz. The specific heat of the samples was measured in a physical property measurement system (PPMS 6000) from Quantum Design through a thermal relaxation method from 2 to 300 K.

**Atomic simulation model in MD simulation**. Typical samples of martensite or strain glass are mimicked by $Zr_{1-x}Ni_x$ model alloys, which contain up to 432,000 atoms. The Ni point defect atoms randomly replace a given atomic concentration $x$ % of the Zr atoms. Then, the systems with randomly distributed Ni atoms were allowed to "age" for 10000 steps of Monte Carlo (MC) simulations[39] with NVT ensemble to achieve the stable solid solution state. A semi-empirical interatomic potential is used to describe the Zr–Ni systems[40]. The potential has been broadly used in previous investigations[41,42] for the BCC-HCP martensitic transformation (MT). Here it should be noted that one should not expect such semi-empirical potential descriptions to capture every detailed aspect of the specific alloys, but such approaches have been widely acceptable for modeling glass transitions and related phenomena. Then, the molecular dynamic (MD) simulations are performed by the annealing of "aged" samples at specified temperatures within the isothermal-isobaric ensemble. The corresponding microstructure or phase structure is identified by order parameter $\eta_i$, which is estimated by the lattice distortion associated with the BCC to HCP transformation[43]. Our previous work has indicated that the properties of strain glass can be well reproduced by the atomistic models of Zr–Ni alloy system[36]. For example, as shown in Supplementary Fig. 5, it well captures the generic temperature-composition phase diagram with the crossover from the ordered martensite to strain glass.

**VDOS and specific heat calculated in simulation**. Using the MD simulation results, we performed isothermal annealing at $T = 40$ K adopting the NVT simulation. With the help of the fluctuation–dissipation theorem, the accumulated data of atomic trajectory from MD simulations can be used to calculate the dynamical matrix, phonon spectrum and VDOS. A detailed description of this method can be found in ref. [44].

Recent works have described the strain glass as isolated martensite nanodomains embedded in the parent phase matrix with the help of the dopants induced strain networks[36]. Here, the atomic pinning method is adopted to extract the VDOS of the nanodomain regions and parent phase matrix separately. For each case, we only allow the atoms within the concerned region to relax while freeze the other when carrying out MD simulations. Then, the atomic displacement within the unpinned regions is used to calculate the VDOS using two-point correlations.

The corresponding heat capacities are calculated based on the harmonic approximation of the free energy. In harmonic approximation, the free energy of the STG system can be estimated from the vibrational density of states $g(\omega)$ directly. Thus, we can obtain the molar heat capacity as

$$c(T) = 3N_A k_B \int_0^\infty \left(\frac{\hbar\omega}{2k_B T}\right)^2 \sinh^{-2}\left(\frac{\hbar\omega}{2k_B T}\right) g(\omega) d\omega,$$

where $k_B$ is Boltzmann's constant, $N_A$ is Avogadro's number, $\omega$ is the phonon frequency, and $g(\omega)$ is the normalized VDOS.

**Transverse dynamical structure factors calculated in simulation**. In order to determine the Ioffe–Regel limit of TA branches, we calculated the transverse dynamical structure factors $S_T(\mathbf{k}, \omega)$ of proper directions. We adopted the Dynasor package for the efficient calculation of dynamical structure factors from MD trajectories[45]. The transverse dynamical structure factor, $S_T(\mathbf{k}, \omega)$, can be estimated by the particle velocity correlation function, $C_T(\mathbf{k}, t)$. Specifically, here we defined a current density $\mathbf{j}(\mathbf{r}, t)$ and $\mathbf{j}(\mathbf{k}, t)$ as

$$\mathbf{j}(\mathbf{r}, t) = \sum_i^N \mathbf{v}_i(t)\delta(\mathbf{r} - \mathbf{r_i}(t)),$$

$$\mathbf{j}(\mathbf{k}, t) = \sum_i^N \mathbf{v}_i(t)e^{i\mathbf{k}\cdot\mathbf{r_i}(t)},$$

where $N$ is the number of particles, $\mathbf{v}_i(t)$ is the velocity of particle $i$ at time $t$, and $\mathbf{r}_i(t)$ is the position of particle $i$ at time $t$. This density can be split into a longitudinal and transverse part. The corresponding transverse part can be extracted by

$$\mathbf{j}_T(\mathbf{k}, t) = \sum_i^N \left[\mathbf{v}_i(t) - \left(\mathbf{v}_i(t) \cdot \hat{\mathbf{k}}\right)\hat{\mathbf{k}}\right]e^{i\mathbf{k}\cdot\mathbf{r_i}(t)},$$

where $\hat{\mathbf{k}}$ denotes the unit vector. Then, the correlation functions, $C_T(\mathbf{k}, t)$, can be calculated by

$$C_T(\mathbf{k}, t) = \frac{1}{N}\langle \mathbf{j}_T(\mathbf{k}, t) \cdot \mathbf{j}_T(-\mathbf{k}, 0)\rangle.$$

With the help of fast Fourier transformation, the correlation functions $C_T(\mathbf{k}, t)$ can be transformed to the frequency domain, i.e., $C_T(\mathbf{k}, \omega)$. The frequency dependent dynamical structural factor $S_T(\mathbf{k}, \omega)$ is then given as

$$S_T(\mathbf{k}, \omega) = k^2 C_T(\mathbf{k}, \omega)/\omega^2.$$

Here, we used a smaller model of Zr-5%Ni alloy, which contains up to 54,000 atoms. To improve the statistics, The protocol was repeated by using 10 independent equilibrium models, and these ten configurations were averaged to estimate $S_T(\mathbf{k}, \omega)$.

## Data availability
The datasets that support the findings of this study are provided in the Source Data file. Source data are provided with this paper.

## Code availability
MD simulations were performed using the LAMMPS software package, which is available at https://lammps.sandia.gov. The dynamic structure factor calculation employed in this work is available at https://dynasor.materialsmodeling.org/.

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

## Acknowledgements
The authors are thankful to H.Y. Bai, B.S. Shang, and H.P. Zhang for helpful discussions. This research was supported by the National Natural Science Foundation of China (No. 51901243, 61888102, 11790291, 51931004, 51621063, 51971238, and 51871177), China Postdoctoral Science Foundation (No. 2019M650880), Natural Science Foundation of Guangdong Province (2019B030302010), the Strategic Priority Research Program of the Chinese Academy of Sciences (XDB30000000) and 111 project 2.0 (BP2018008).

## Author contributions
W.H.W. and X.D.D. supervised the project. S.R. conceived the idea, designed and performed the experiments on the DSC, DMA, and PPMS. H.X.Z. and X.F.T. designed and performed the MD simulations. S.R., H.X.Z., X.D.D. and W.H.W. discussed and analyzed the data and wrote the manuscript. S.R. and H.X.Z. contributed equally to this work. Y.H.S., B.A.S. and D.Z.X. participated in data interpretation and manuscript correction. All authors contributed to comment on the manuscript writing and the result discussions.

## Competing interests
The authors declare no competing interests.

**Additional information**

