## [Peer Review File · Nature Communications]

REVIEWER COMMENTS

Reviewer #1 (Remarks to the Author):

The paper claims that a boson peak in measured heat capacity data of a strain glass originates with soft vibrations occurring around nanodomains, rather than within them. The data by itself cannot distinguish whether the softening comes from a damping of phonons, which is expected in glasses, or a softening of the collective phonon excitations. Previous experiments using neutron scattering on strain glasses (Ref. 33) reveal a damping of the transverse acoustic (TA) phonons, but no softening of the collective phonon mode frequencies. This damping could explain the boson peak in the heat capacity. Furthermore, the damping occurs at temperatures above the glass temperature, demonstrating that it does not require the existence of the nanodomains. Hence, previous experimental results contradict the interpretation given in the manuscript, although the new experimental results are consistent with what is known since the boson peak is expected with damping of the TA phonon.

The discussion regarding vibrations around the nanodomains is not clear. Phonons are collective excitations that necessarily must extend periodically over some distance in the crystal in order to be defined as phonons (this is required if the Fourier transforms from real space and time to momentum-energy space are to produce sharp dispersion curves). If vibrations are only altered locally then these are not collective modes, or phonons. Rather, if the vibrational properties are locally different at low energies this is expected to scatter the long wavelength phonons. Strongly scattering phonons are damped phonons. Hence, what they are describing in this paper would seem to indicate a damping of the phonons if viewed properly in reciprocal space. Yet they claim that their interpretation conflicts with the direct observations of damping in the phonons.

I do not recommend publishing this manuscript in its current form. The measurements appear correct but the interpretation needs work. For example, the authors might consider calculating momentum resolved spectra functions from the simulations. This would enable them to make a more meaningful distinction between phonon damping and phonon softening, and also make direct comparisons with neutron scattering measurements. Given the description of their model, I believe that they will find that distinguishing damping and low energy vibrations is not so simple and requires a more nuanced discussion at least.

Reviewer #2 (Remarks to the Author):

Authors observed a hump in specific heat (C_p) divided by cubed temperature for a crystalline alloy. They conducted molecular-dynamic simulations, and showed that the anomaly in C_p corresponds to a peak of low-frequency vibrational states that stem from softening of the transverse acoustic (TA) branches somewhere around a centre of Brillouin zone.

Altogether, this looks as a very accurate and quite professional research, with clearly reliable and novel results.

However, despite of my deep appreciation of the research, I do not see that the manuscript merits publishing in Nature Communication. The reason is a clearly misleading identification of the observed phenomenon with an excess of low-energy states in glasses above the level of acoustic states expected from Debye law, i.e., with the so-called boson peak.

As authors show, the discussed anomaly of C_p is explained by flattening of transverse acoustic branches. Therefore, identifying a non-Debye behaviour of dispersion relations, authors should not expect Debye behaviour of DOS in the relevant energy range.

Thus, the observed peaks in C_p/T^3 and in reduced DOS are not the anomalous (or “relaxational”) modes above the expected Debye level. On the contrary, they are expected deviations from Debye behaviour suggested by dispersion relations.

Aleksandr Chumakov

Report on NCOMMS-20-40100

General comments

The authors report on the experimental identification of a boson-peak (BP) anomaly in the specific heat of certain compounds in the strain glass phase. A BP is clearly seen around 10K in the Debye normalized specific heat and it is supported by simulation studies. Moreover, the BP anomaly is associated to the softening of transverse acoustic phonons, which is indicated as the underlying physical reason for this feature. The paper presents interesting results but cannot be accepted in Nature Communications in its present form. The authors are strongly encouraged to prepare a revised manuscript with major revisions, following the lines presented in detail below.

Relevance of the paper

This paper is timely for various reasons:

- (I) It provides yet another confirmation that the BP anomaly can appear also in ordered structure where disorder cannot be the mechanism behind it. This is a very relevant point of discussion over the past few years. The BP anomaly has been systematically observed in non-disordered systems, atomic and molecular cryocrystals and halomethanes, organic crystals etc [7, 3, 5] and in quasicrystals/incommensurate structures [6]. This clearly indicates that the BP is not universally given by disorder and that most of the existing theories (especially the most famous "heterogeneous elasticity theory" by Schirmacher and Ruocco) fail in these cases.
- (II) This experimental result is another proof that the idea of the BP being a relic of the Van-Hove singularity is plain wrong, since it shows a clear separation of scales between the two. The presented evidence nicely align with other evidences reported by other authors recently.
- (III) The experimental data, and also the interpretation of the authors, support a recent theoretical model [2](Ref.[23] in the manuscript) based on an effective hydrodynamic description of the phonons dynamics in ordered and disordered solids. In particular, a follow-up of such theory presented in [1] can directly explain all the features observed by the authors (I will get back to this point in the next section).

Points to be improved

1. The authors ask the following question : "does strain glass possess other dynamic relaxation modes?" (lines 48-49) and they conclude that the BP is the proof for the existence of such extra mode. I do not agree with this statement. As the authors

show very nicely, the BP does not arise because of the presence of new relaxation modes but simply because of the "renormalization"/softening of the acoustic phonon degrees of freedom. I would say that the conclusion of the paper is exactly the opposite. There are no new relaxation modes but nevertheless a nice boson peak anomaly. Why the presence of a BP should imply an extra relaxation mode and what is that mode? The soft acoustic phonons, and the crossover from phonon to diffusons (Ioffe-Regel crossover) predicted for all crystals, can perfectly explain it. Also, the analysis of the authors is clear, such mode does not exist! I suggest the authors to revisit their conclusions in light of this evidence.

2. In lines 56-57, the authors write that it is widely accepted that the boson peak is a manifestation of structural disorder. This is totally misleading and not correct. Simply the work of the authors show that this is not the case. Moreover, as I already anticipated, there are recently plenty of evidence that the BP is not generically a manifestation of structural disorder. The BP has been observed in several systems with no structural disorder [7, 3, 5, 6]. I suggest the authors modify this wrong statement accordingly. Also Ref.[23] must not be included in the references in line 59, since the scope of that work is exactly to provide a theoretical explanation for the appearance of the BP in systems with no disorder. That is how that paper is now being recognized in the literature, i.e. as the first prediction of BP in ordered systems. It is important to re-write that part and be more clear about the current theoretical dispute mentioning the proper references (which I partially indicated above) and distinguishing models with disorder and models without. In particular Ref. [23] should be clearly acknowledged as the first paper that presents a theory of BP in ordered crystalline systems.

Figure 1: Phonon softening in the theoretical model of [1].

3. A very important step provided by this work is the identification of the fundamental physical reason behind the BP anomaly in systems with no structural disorder.

Figure 2: The results from the theoretical model of [1]. They appear to be in full-agreement with the analysis by the authors.

Nevertheless, the theoretical background is quite poor and the comparison with the existing theoretical frameworks is absent. As I will explain in the following, the authors analysis is in full agreement with the theory of [1], which is a follow-up of Ref. [23]. In [1], the dispersion relation of the acoustic phonons can be found by solving the equation:

$$\omega^2 - v^2 k^2 + A k^4 + i \omega D k^2 = 0 \quad (1)$$

Focusing on the real part of the dispersion relation, i.e. the energy of the acoustic phonons (which correspond with what is shown in Fig.4 of the authors manuscript), one finds:

$$\omega = \sqrt{v^2 k^2 - A k^4} \quad (2)$$

where v is the phonon speed given, as usual, in terms of the elastic moduli and A an effective parameter representing anharmonic effects in the system. Upon specializing to transverse acoustic phonons, from Eq.(2) is clear that phonon softening can be realized in two separate ways: (i) by decreasing the speed v and/or (ii) by increasing the anharmonic term A . This is indeed shown in Fig.1.

Using standard methods, one can proceed with the derivation of the density of states and the specific heat corresponding to such acoustic phonons. The model shows a clear BP whose intensity grows with the phonon softening and which is well-separated from the corresponding Van-Hove singularity. See Fig.2.

In summary, it is evident that the theoretical framework of Refs. [2, 1] beautifully matches with the experimental and simulation analysis done by the authors. Therefore, I would suggest to stress more this point as a nice marriage between experiment and theory and the opening of a totally new conception of glassy-like anomalies in absence of structural disorder.

This discussion will give the paper the allure that is required for Nature Communications, whereas the paper in its current version lacks the type of focus and message

needed for Nature Communications. For example, the authors could present some of the plots that I showed above which clearly explain the observed phenomena. Without this more quantitative explanation of the data in terms of current theories, the bare experimental data may provide a misleading or confused message to the community.

4. Following on this direction, I think the authors should be more precise about the importance and the main messages of their results. (I) The theoretical models relying on disorder cannot explain the BP in this system, (II) the idea of the BP being the VH is clearly falsified by these experimental results (as written by the authors in line 152), (III) the results clearly support the theoretical model of [2, 1] as a viable explanation for the BP in this system. These three points must be clearly stated in the abstract, since are the major implications of the authors analysis.
5. In the context of models which predict a BP from phonon softening (i.e. softening of shear modulus) in both glasses and perfectly ordered non-centrosymmetric crystals, and crystals with locally non-centrosymmetric nearest-neighbour force imbalance, also Ref. [4] should be mentioned and discussed. This reference also showed that the van Hove singularity is distinct from the BP in crystals.

Final evaluation

In my opinion, the manuscript represents a solid piece of work with a potential impact both from the experimental but also from the theoretical/conceptual side, provided the authors make an effort to offer a more quantitative interpretation of their results, as explained above. The work constitutes potentially strong evidence towards a complete re-thinking process about glass-like anomalies in solids, and it is an experimental proof that almost all the existing theories for BP anomalies fail in capturing its origin. It moreover appears in nice agreement with recent theoretical progress opening a new path towards the understanding of glassy anomalies in ordered phases.

I am happy to reconsider this manuscript for publication in Nature Communications after important revisions regarding the points above are made.

References

- [1] M. Baggioli and A. Zaccone. Unified theory of vibrational spectra in hard amorphous materials. *Phys. Rev. Research*, 2:013267, Mar 2020.
- [2] Matteo Baggioli and Alessio Zaccone. Universal origin of boson peak vibrational anomalies in ordered crystals and in amorphous materials. *Phys. Rev. Lett.*, 122:145501, Apr 2019.

- [3] J. F. Gebbia, M. A. Ramos, D. Szewczyk, A. Jezowski, A. I. Krivchikov, Y. V. Horbatenko, T. Guidi, F. J. Bermejo, and J. Ll. Tamarit. Glassy anomalies in the low-temperature thermal properties of a minimally disordered crystalline solid. *Phys. Rev. Lett.*, 119:215506, Nov 2017.
- [4] R. Milkus and A. Zaccone. Local inversion-symmetry breaking controls the boson peak in glasses and crystals. *Phys. Rev. B*, 93:094204, Mar 2016.
- [5] Manuel Moratalla, Jonathan F. Gebbia, Miguel Angel Ramos, Luis Carlos Pardo, Sanghamitra Mukhopadhyay, Svemir Rudić, Felix Fernandez-Alonso, Francisco Javier Bermejo, and Josep Lluís Tamarit. Emergence of glassy features in halomethane crystals. *Phys. Rev. B*, 99:024301, Jan 2019.
- [6] G. Reményi, S. Sahling, K. Biljaković, D. Starešinić, J.-C. Lasjaunias, J. E. Lorenzo, P. Monceau, and A. Cano. Incommensurate systems as model compounds for disorder revealing low-temperature glasslike behavior. *Phys. Rev. Lett.*, 114:195502, May 2015.
- [7] M. A. Strzhemechny, A. I. Krivchikov, and A. Jeowski. Heat capacity of molecular solids: The special case of cryocrystals. *Low Temperature Physics*, 45(12):1290–1295, 2019.

Responses to Reviewers' Reports on NCOMMS-20-40100

We thank all the three reviewers for their careful review of this manuscript. We are pleased that all the three reviewers raise positive comments on this work, especially on the experimental part, and provide constructive suggestions to help us substantially improve the quality of this work. According to the editors' suggestions, we made major revisions to the manuscript to solve the main concerns of the reviewers, which include (1) the discrimination of the role of phonon softening and phonon damping in the BP-like anomaly, (2) the improper identification of the BP(-like) anomaly in strain glass and (3) the improvement of the theoretical background. The additional simulation work and literature research can solve these main concerns, and other minor questions are also addressed.

Regarding the demand to distinguish the role of phonon softening and phonon damping by Reviewer 1, we followed his/her good comments and performed additional simulation work to get the momentum resolved spectra functions. New simulation results show that the phonon damping is not strong enough to achieve the Ioffe-Regel limit in the first Brillouin zone for the [001] transverse acoustic branch of the strain glass, in which branch the phonon softening takes place. It means that the vibrations contributing to the BP in strain glass are still well-defined phonons, which is a critical piece of evidence to support that the boson peak in strain glass is attributed to the phonon softening, rather than the phonon damping. We thank the reviewer for the crucial suggestions that help us to develop a deeper insight into this work.

Regarding the improper identification raised by Reviewer 2, we are grateful for his deep appreciation of this work, and his agreement on the phonon softening as the origin of the BP-like anomaly in strain glass. The main concern can be properly addressed according to the following reasons. On one hand, some recent work has shown that the anomaly similar to the BP in structural glasses exists in some crystalline systems with less or even no disorder. They all are identified as the boson peak, and so is it in this work. On the other hand, in order to explain the BP in these crystalline systems, it is of importance to develop new understanding of the origin of the BP in solids, since the prevailing models in structural glasses generally rely on the structural disorder and are not applicable to the ordered systems. This work reveals that the BP-like anomaly can be generated by the phonon softening. The mechanism itself is something new in this work, and no research has reported the mechanism related to TA phonon softening so far. Therefore, the new mechanism related to phonon softening may lead to a re-thinking of the BP behavior in solids both with disorder and without, as pointed out by Reviewer 3. We thank the reviewer for guiding us to sharpen our key findings and improve the revised manuscript.

As for the improvement of the theoretical background, we deeply thank Reviewer 3 for the positive assessment of this work and for kindly providing detailed suggestions that help us deepen the understanding of the origin of the BP and significantly improve the quality of this work. According to his/her helpful comments, we studied the frequency dispersion of different compositions and found there is a nice match between our simulation results and the theoretical model pointed out by the reviewer, which lends further support to the main conclusion of this work — the BP-like anomaly of strain glass stems from the phonon softening. We deeply acknowledge all the suggestions of the reviewer and follow them to prepare a revised

manuscript with major revisions.

In summary, all the concerns have been properly addressed, as shown below as well as in the revised manuscript. We acknowledge all the three reviewers again for their helpful comments and suggestions which help us to sharpen our key findings and lead to the substantial improvement of the quality of the revised manuscript.

Response to Reviewer 1

Comment: *The paper claims that a boson peak in measured heat capacity data of a strain glass originates with soft vibrations occurring around nanodomains, rather than within them. The data by itself cannot distinguish whether the softening comes from a damping of phonons, which is expected in glasses, or a softening of the collective phonon excitations. Previous experiments using neutron scattering on strain glasses (Ref. 33) reveal a damping of the transverse acoustic (TA) phonons, but no softening of the collective phonon mode frequencies. This damping could explain the boson peak in the heat capacity. Furthermore, the damping occurs at temperatures above the glass temperature, demonstrating that it does not require the existence of the nanodomains. Hence, previous experimental results contradict the interpretation given in the manuscript, although the new experimental results are consistent with what is known since the boson peak is expected with damping of the TA phonon.*

The discussion regarding vibrations around the nanodomains is not clear. Phonons are collective excitations that necessarily must extend periodically over some distance in the crystal in order to be defined as phonons (this is required if the Fourier transforms from real space and time to momentum-energy space are to produce sharp dispersion curves). If vibrations are only altered locally then these are not collective modes, or phonons. Rather, if the vibrational properties are locally different at low energies this is expected to scatter the long wavelength phonons. Strongly scattering phonons are damped phonons. Hence, what they are describing in this paper would seem to indicate a damping of the phonons if viewed properly in reciprocal space. Yet they claim that their interpretation conflicts with the direct observations of damping in the phonons.

I do not recommend publishing this manuscript in its current form. The measurements appear correct but the interpretation needs work. For example, the authors might consider calculating momentum resolved spectra functions from the simulations. This would enable them to make a more meaningful distinction between phonon damping and phonon softening, and also make direct comparisons with neutron scattering measurements. Given the description of their model, I believe that they will find that distinguishing damping and low energy vibrations is not so simple and requires a more nuanced discussion at least.

Response:

We thank the reviewer for his/her positive comments on the experimental results and very relevant suggestions on the additional simulation work, which help us to largely enhance the validity of this work. The key concern of the reviewer is to determine the exact origin of the BP-like anomaly in strain glass between phonon softening and phonon damping. Another related question is to deal with the conflict between this work and the direct observations of glasslike phonon damping in strain glass [PRL 120, 245701 (2018), i.e., Ref. 33 in the previous

manuscript]. We followed the reviewer's suggestions and calculated the momentum resolved spectra functions. The new simulation data clearly shows that it is the phonon softening that plays the central role in giving rise to the BP-like anomaly, but the phonon damping. The conflict between this work and the literature is also properly addressed. As a result, all the reviewer's concerns can be properly resolved. We revised our manuscript to address these concerns.

As pointed out by the reviewer, the boson peak is generally believed to be caused by the damping of TA phonons in structural glasses. The phonon damping can decrease the mean free path of phonons. When the phonon damping is strong enough, the mean free path of phonons becomes smaller than their wavelength, then these strongly damped lattice waves cease to exist as well-defined phonons. The crossover is known as the Ioffe–Regel limit, at which the mean free path is comparable to the wavelength. It can also be treated equivalently in terms of frequency: above the Ioffe–Regel crossover frequency ω_{IR} , the phonons become ill-defined phonons or damped phonons. It has been demonstrated that the frequency of boson peak ω_{BP} is equivalent to the Ioffe–Regel limit for the transverse acoustic wave ($\omega_{\text{IR-TA}}$) in structural glasses [Nat. Mater. 7, 870 (2008), PRB 87, 134203 (2013)]. As a result, the boson peak frequency is believed to mark the characteristic frequency of transverse vibrational modes, where the transverse phonons transform from propagating to diffusive modes. Therefore, a crucial piece of evidence to distinguish the contribution of the phonon softening and the phonon damping in this work is to discriminate the relation between ω_{BP} and $\omega_{\text{IR-TA}}$ in strain glass.

We thus need to obtain the $\omega_{\text{IR-TA}}$ of two TA branches, namely [110] and [001], because the [110] TA branch corresponds to the basal mode with displacements matching the martensitic transformation (known as the TA_2 mode), in which branch the glasslike phonon damping has been reported (Ref. 33), while the phonon softening corresponding to ω_{BP} takes place in the [001] branch. We adopted the calculation method in the reference [Nat. Mater. 7, 870 (2008)] to calculate the $\omega_{\text{IR-TA}}$ in these two branches. Figures R1-1(a-b) exhibit the transverse dynamical structure factors $S_T(k, \omega)$ along [110] and [001] at 40K for Zr-5%Ni strain glass respectively. The solid lines are the fits by the damped harmonic oscillator model according to the reference [Nat. Mater. 7, 870 (2008)], which is

$$S_T(k, \omega) = \frac{A_T(k)}{[\omega^2 - \Omega_T^2(k)]^2 + \omega^2 \Gamma_T^2(k)},$$

where $\Omega_T(k)$ corresponds to the excitation frequency, $\Gamma_T(k)$ corresponds to the full-width at half-maximum of the excitations or the phonon damping coefficient, and $A_T(k)$ is the fitting coefficient. The obtained $\Omega_T(k)$ and $\Gamma_T(k)$ of different directions are plotted in Figs. R1-1(c-d) respectively. In general, $\Omega_T(k)$ obeys a linear dispersion relation with k , and $\Gamma_T(k)$ follows a k^2 law, which is consistent with the literature [Nat. Mater. 7, 870 (2008) and references therein]. The Ioffe-Regel limit condition is given by $\Omega_T(k) = \pi \Gamma_T(k)$, which corresponds to the intersection point of two curves in Figs. R1-1(c-d).

It is noted that a very low $\omega_{\text{IR-TA2}}$ (~ 0.4 THz) is observed along [110] in Fig. R1-1(c), which corresponds to the TA_2 mode. It means that strong phonon damping occurs in the TA_2 mode, and this result is consistent with the experimental observation in previous work (Ref. 33). More importantly, there is no intersection point in the first Brillouin zone along [001] in Fig. R1-1(d), which corresponds to the soft mode but not associated with the transition displacements. The

absence of the intersection point in the [001] TA branch means that the lattice waves are still well-defined phonons in the first Brillouin zone of this direction, and thus ω_{BP} is unrelated to ω_{IR-TA} along [001]. Therefore, it clearly shows that the BP-like anomaly is caused by the well-defined phonons and unrelated to the phonon damping.

Fig. R1-1. Transverse phonon dynamics and the dispersion relations along [110] and [001]. **a-b**, Transverse dynamical structure factors of the strain glass Zr-5%Ni along (a) [110] and (b) [001], respectively. Solid lines are the fits by the damped harmonic oscillator model. **c-d**, Dispersion relation $\Omega_T(k)$ and excitation broadening $\pi\Gamma_T(k)$ for transverse motion along (c) [110] and (d) [001] of 5Ni, respectively.

Moreover, our new simulation results of the two branches are actually both consistent with the experimental results in the reference (Ref. 33). In that reference, scientists reported that the glasslike phonon damping takes place in the TA mode with displacements matching the martensitic transformation (i.e., the TA₂ mode), while no TA broadening is found along the directions not associated with the transition displacements. We draw the contour maps of Fig. R1-2 according to Figs. R1-1(c-d), which clearly exhibit strong phonon damping in the TA₂ mode in Fig. R1-2(a), while the phonon broadening is very weak in the [001] TA branch in Fig. R1-2(b). Therefore, there is no conflict anymore after examining the phonon damping of these two different directions.

At last, there is a minor question left regarding the comment that *the damping occurs at temperatures above the glass temperature, demonstrating that it does not require the existence of the nanodomains.* In fact, the nanodomains are observed by HRTEM at the temperature well above the glass temperature T_g ($\sim T_g + 100$ K) [PRL 112, 025701 (2014)], and the nanodomains are expected to appear at the temperature even 200 K higher than T_g , according to the electrical resistivity results [PRB 81, 224102 (2010)]. Therefore, we think the nanodomains may already exist at the temperature where the phonon damping occurs in the reference (Ref. 33).

To sum up, we followed the very relevant comments of the reviewer and calculated $\omega_{\text{IR-TA}}$ of different directions in strain glass. New simulation results exhibit that strong phonon damping occurs at the TA_2 mode along $[110]$, which leads to a very low $\omega_{\text{IR-TA}_2}$, consistent with the result in the literature, while ω_{BP} corresponds to the frequency of phonon softening along $[001]$ and is uncorrelated to $\omega_{\text{IR-TA}}$. Thus, the BP-like anomaly indeed stems from the phonon softening and is unrelated to the phonon damping. Then all the reviewer's concerns are properly resolved. We acknowledge the reviewer for his/her crucial suggestions that help us to develop a much deeper insight into this work.

Fig. R1-2. Contour maps of the normalized dynamical structure factor on the plane (ω, k) for directions (a) $[110]$ and (b) $[001]$, respectively. The normalized factor is obtained by dividing by the maximum of curves at each fixed k . The black full line corresponds to the dispersion relation fitted by the damped harmonic oscillator model. The white horizontal dash in (a) shows the transverse Ioffe-Regel crossover frequency $\omega_{\text{IR-TA}_2}$ along $[110]$.

Changes made in the revised manuscript:

1. Page 27, Line 621:
We added Fig. R1-1 and Fig. R1-2 into the revised manuscript as Fig. 5.
2. Page 9, Line 245:
We revised the discussion part and provided a detailed description of Fig. 5.

Response to Reviewer 2

Comment: *Authors observed a hump in specific heat (C_p) divided by cubed temperature for a crystalline alloy. They conducted molecular-dynamic simulations, and showed that the anomaly in C_p corresponds to a peak of low-frequency vibrational states that stem from softening of the transverse acoustic (TA) branches somewhere around a centre of Brillouin zone.*

Altogether, this looks as a very accurate and quite professional research, with clearly reliable and novel results.

However, despite of my deep appreciation of the research, I do not see that the manuscript merits publishing in Nature Communication. The reason is a clearly misleading identification of the observed phenomenon with an excess of low-energy states in glasses above the level of acoustic states expected from Debye law, i.e., with the so-called boson peak.

As authors show, the discussed anomaly of C_p is explained by flattening of transverse acoustic branches. Therefore, identifying a non-Debye behaviour of dispersion relations, authors should not expect Debye behaviour of DOS in the relevant energy range.

Thus, the observed peaks in C_p/T^3 and in reduced DOS are not the anomalous (or “relaxational”) modes above the expected Debye level. On the contrary, they are expected deviations from Debye behaviour suggested by dispersion relations.

Aleksandr Chumakov

Response:

We deeply thank the reviewer for considering this work as *a very accurate and professional research with reliable and novel results* and are grateful for his agreement on phonon softening as the origin of the C_p anomaly in strain glass. The main concern of the reviewer is whether it is proper to identify the anomaly in the low-temperature C_p of strain glass as the boson peak in structural glasses, since the underlying mechanism in strain glass is the flattening of transverse acoustic (TA) branches, different from the understanding of the origin of the BP in structural glasses. In order to solve this concern, we performed detailed literature research and additional simulation work, and these new data may properly resolve the reviewer’s concern.

As pointed out by the reviewer, the BP corresponds to an excess of low-energy states above the expected Debye level, which is manifested by a peak in the reduced VDOS at low frequencies (~ 1 THz) or a hump in the C_p at low temperatures (~ 10 K). The BP is generally considered as the unique characteristic of structural glasses as compared with their crystalline counterparts. However, there are emerging reports recently exhibiting a similar anomaly in some crystalline systems with less or even no disorder, such as organic crystals and atomic/molecular cryocrystals, which are generally identified as the BP [PRL 119, 215506 (2017), PRB 99, 024301 (2019), Low Temp. Phys. 45, 1290 (2019)]. In this work, we reported that Ti-Ni binary strain-glass-involved crystalline alloys exhibit characteristics similar to the BP of metallic glass in both C_p and reduced VDOS. Thus, we think it is proper to follow the above literature and identify this anomaly as the BP-like anomaly.

On the other hand, since the prevailing theoretical models in structural glasses generally rely on the assumption of the existence of structural disorder in the system, it is clear that the BP in these crystalline systems cannot be interpreted by these models. Therefore, it is of importance to develop new understanding of the origin of the BP to explain the observations of the BP in these crystalline systems. The key point of this work is that we found the BP-like anomaly in strain glass stems from a new mechanism related to TA phonon softening. Thus, this new mechanism may provide a new perspective for understanding the origin of the BP.

We are glad that the reviewer agrees with the mechanism of phonon softening as the origin of the BP-like anomaly. Meanwhile, we notice there are some misunderstandings from the reviewer regarding the comment that *the observed peaks in C_p/T^3 and in reduced DOS are not the anomalous (or “relaxational”) modes above the expected Debye level. On the contrary, they are expected deviations from Debye behaviour suggested by a non-Debye behaviour of dispersion relations.* We obtain two meanings from this comment.

First, this comment focuses on the link between the observed peaks and the corresponding dispersion relation, namely, the reviewer believes that phonon softening, as a non-Debye behavior of dispersion relations, will inevitably lead to expected deviations from Debye behavior. However, this link is not so explicit, since phonon damping is also observed in strain glass experimentally [PRL 120, 245701 (2018)], and we did not rule out the possibility that phonon damping is responsible for the BP-like anomaly in strain glass in the previous manuscript, as pointed out by Reviewer 1. Our new simulation results further show that the BP is unrelated to the phonon damping, and is attributed to the phonon softening. More importantly, we need to emphasize that the phonon softening itself is something new in this work. So far as we know, no research has reported that a boson peak can be caused by a mechanism related to TA phonon softening, and it is also the first time to find the presence of phonon softening in strain glass at low temperatures far below the glass transition temperature. In comparison, such phonon softening is absent in the martensitic phase (the long-range ordered counterpart of strain glass), and the martensitic phase only exhibits a normal Debye behavior. Moreover, our mechanism is also verified by the theoretical framework pointed out by Reviewer 3. As a result, there is a nice match between our simulation and the theory. Thus, as commented by Reviewer 3, the new mechanism of phonon softening underlying the BP-like anomaly in strain glass may lead to a re-thinking of the BP behavior in solids with disorder and without.

Second, we notice the reviewer comments that *the observed peaks are not the anomalous relaxational modes*, and Reviewer 3 also points out that the statement of “relaxation mode” is misleading, so we realize it was a misjudgment, and we regret that our statement cause confusion to the two reviewers. Therefore, we have clarified that the BP-like anomaly is a new glassy feature in strain glass, rather than a relaxation mode, in the revised manuscript.

To sum up, the reviewer’s concern about the improper identification reflects several weaknesses of the previous manuscript, such as (1) incomplete research background without introducing the BP in systems with less or even no disorder and (2) limited theoretical depth without distinguishing the different roles of phonon softening and phonon damping and providing a detailed discussion about the comparison of this work with the theoretical models with structural disorder and those without. We thank the reviewer for his concern that guides us to make up for these weaknesses in the revised manuscript and rewrite a well-improved version.

Changes made in the revised manuscript:

1. Page 3, Line 63:

We rewrote the introduction to the boson peak. In particular, we added the description of the boson peak in ordered systems, “On the other hand, there are emerging reports recently exhibiting a similar boson peak anomaly in some crystalline systems with less or even no disorder, such as atomic/molecular cryocrystals²⁵, halomethanes²⁶, organic crystals²⁷, and quasicrystals/incommensurate systems²⁸.”

2. Page 9, Line 245:

We added a detailed discussion to compare this work with the theoretical models with disorder and without.

Response to Reviewer 3

Comment 1:

General comments

The authors report on the experimental identification of a boson-peak (BP) anomaly in the specific heat of certain compounds in the strain glass phase. A BP is clearly seen around 10K in the Debye normalized specific heat and it is supported by simulation studies. Moreover, the BP anomaly is associated to the softening of transverse acoustic phonons, which is indicated as the underlying physical reason for this feature. The paper presents interesting results but cannot be accepted in Nature Communications in its present form. The authors are strongly encouraged to prepare a revised manuscript with major revisions, following the lines presented in detail below.

Relevance of the paper

This paper is timely for various reasons:

- (I) It provides yet another confirmation that the BP anomaly can appear also in ordered structure where disorder cannot be the mechanism behind it. This is a very relevant point of discussion over the past few years. The BP anomaly has been systematically observed in non-disordered systems, atomic and molecular cryocrystals and halomethanes, organic crystals etc [7, 3, 5] and in quasicrystals/incommensurate structures [6]. This clearly indicates that the BP is not universally given by disorder and that most of the existing theories (especially the most famous "heterogeneous elasticity theory" by Schirmacher and Ruocco) fail in these cases.*
- (II) This experimental result is another proof that the idea of the BP being a relic of the Van-Hove singularity is plain wrong, since it shows a clear separation of scales between the two. The presented evidence nicely align with other evidences reported by other authors recently.*
- (III) The experimental data, and also the interpretation of the authors, support a recent theoretical model [2] (Ref.[23] in the manuscript) based on an effective hydrodynamic description of the phonons dynamics in ordered and disordered solids. In particular, a follow-up of such theory presented in [1] can directly explain all the features observed by the authors (I will get back to this point in the next section).*

Response:

We are very pleased and deeply grateful that the reviewer not only completely understands the novelty and importance of this work, but also provides a much deeper insight into the origin of the boson peak for us. The reviewer's encouraging comments and constructive suggestions help us to develop a much deeper understanding of the underlying physics of the boson peak in both strain glass and other solids with disorder or without, and thus lead to a substantial improvement of the quality of this work. We made major revisions to the manuscript according to these helpful comments and suggestions.

Comment 2:

Points to be improved

- 1. The authors ask the following question: "does strain glass possess other dynamic relaxation modes?" (lines 48-49) and they conclude that the BP is the proof for the existence of such extra mode. I do not agree with this statement. As the authors show very nicely, the BP does not arise*

because of the presence of new relaxation modes but simply because of the "renormalization"/softening of the acoustic phonon degrees of freedom. I would say that the conclusion of the paper is exactly the opposite. There are no new relaxation modes but nevertheless a nice boson peak anomaly. Why the presence of a BP should imply an extra relaxation mode and what is that mode? The soft acoustic phonons, and the crossover from phonon to diffusons (Ioffe-Regel crossover) predicted for all crystals, can perfectly explain it. Also, the analysis of the authors is clear, such mode does not exist! I suggest the authors to revisit their conclusions in light of this evidence.

Response:

We thank the reviewer for this comment. In the previous version, we considered that it is the first time to find the boson peak in strain glass, so we used the terms “other dynamic relaxation modes” and “new relaxation modes” to refer to the BP anomaly in strain glass. In fact, we agree with the reviewer’s opinion that *there are no new relaxation modes but nevertheless a nice boson peak anomaly*. We realize that the statement of “relaxation mode” is misleading according to this comment. Thus, to avoid confusion, we have modified the statement by removing the two above terms and replacing the term “relaxation mode” with “glassy (dynamic) feature” in the context where the BP in strain glass needs to be referred to.

Comment 3:

- In lines 56-57, the authors write that it is widely accepted that the boson peak is a manifestation of structural disorder. This is totally misleading and not correct. Simply the work of the authors show that this is not the case. Moreover, as I already anticipated, there are recently plenty of evidence that the BP is not generically a manifestation of structural disorder. The BP has been observed in several systems with no structural disorder [7, 3, 5, 6]. I suggest the authors modify this wrong statement accordingly. Also Ref. [23] must not be included in the references in line 59, since the scope of that work is exactly to provide a theoretical explanation for the appearance of the BP in systems with no disorder. That is how that paper is now being recognized in the literature, i.e. as the first prediction of BP in ordered systems. It is important to re-write that part and be more clear about the current theoretical dispute mentioning the proper references (which I partially indicated above) and distinguishing models with disorder and models without. In particular Ref. [23] should be clearly acknowledged as the first paper that presents a theory of BP in ordered crystalline systems.*

Response:

We thank the reviewer for this comment. We agree with the reviewer that *the BP is not generically a manifestation of structural disorder*, and we should have made this point more explicit. According to this comment, we rewrite this part by introducing the theoretical dispute more clearly, mentioning the BP in systems with no structural disorder, and distinguishing the models with disorder and models without. Especially, we emphasize that Ref. [23] [Phys. Rev. Lett. **122**, 145501 (2019)] is a very important achievement in this field, which provides the first theoretical framework to successfully explain the origin of the boson peak in the systems with less or no disorder.

Changes made in the revised manuscript:

1. Page 2, Line 57:

We rewrote the introduction part by (1) introducing the main theoretical models in structural glasses, (2) pointing out the emergence of BP in systems with less or no structural disorder, and (3) emphasizing that the reference [Phys. Rev. Lett. **122**, 145501 (2019)] provides the first theoretical explanation for the boson peak in these systems with less or no disorder.

Comment 4:

Figure 1: Phonon softening in the theoretical model of [1].

3. A very important step provided by this work is the identification of the fundamental physical reason behind the BP anomaly in systems with no structural disorder.

Figure 2: The results from the theoretical model of [1]. They appear to be in full-agreement with the analysis by the authors.

Nevertheless, the theoretical background is quite poor and the comparison with the existing theoretical frameworks is absent. As I will explain in the following, the authors analysis is in full agreement with the theory of [1], which is a follow-up of Ref. [23]. In [1], the dispersion relation of the acoustic phonons can be found by solving the equation:

$$\omega^2 - v^2 k^2 + Ak^4 + i\omega Dk^2 = 0 \tag{1}$$

Focusing on the real part of the dispersion relation, i.e. the energy of the acoustic phonons (which correspond with what is shown in Fig.4 of the authors manuscript), one finds:

$$\omega = \sqrt{v^2 k^2 - Ak^4} \tag{2}$$

where v is the phonon speed given, as usual, in terms of the elastic moduli and A an effective parameter representing anharmonic effects in the system. Upon specializing to transverse acoustic phonons, from Eq. (2) is clear that phonon softening can be realized in two separate ways: (i) by decreasing the speed v and/or (ii) by increasing the anharmonic term A . This is

indeed shown in Fig.1.

Using standard methods, one can proceed with the derivation of the density of states and the specific heat corresponding to such acoustic phonons. The model shows a clear BP whose intensity grows with the phonon softening and which is well-separated from the corresponding Van-Hove singularity. See Fig.2.

In summary, it is evident that the theoretical framework of Refs. [2, 1] beautifully matches with the experimental and simulation analysis done by the authors. Therefore, I would suggest to stress more this point as a nice marriage between experiment and theory and the opening of a totally new conception of glassy-like anomalies in absence of structural disorder.

This discussion will give the paper the allure that is required for Nature Communications, whereas the paper in its current version lacks the type of focus and message needed for Nature Communications. For example, the authors could present some of the plots that I showed above which clearly explain the observed phenomena. Without this more quantitative explanation of the data in terms of current theories, the bare experimental data may provide a misleading or confused message to the community.

Response:

We acknowledge the reviewer for this very important comment which helps us to develop a much deeper understanding of the origin of the boson peak. We performed a careful literature investigation according to the reviewer's suggestion, and we not only added a detailed theoretical background in the revised manuscript but made a clear comparison with the existing theoretical frameworks as well.

We adopt Equation (2) proposed by the reviewer as the theoretical basis. Meanwhile, because our simulation results exhibit that the phonon softening along [001] occurs somewhere around the center of the Brillouin zone, as shown in Fig. R3-1(a), we need to add an additional higher term into the TA dispersion relations to fit this characteristic. Thus, the equation is written as,

$$\omega^2(k) = v_T^2 k^2 - A_T k^4 + B_T k^6, \quad (3-1)$$

where v_T is the phonon speed of propagation of the TA branch and A_T is an effective parameter representing anharmonic effects of the TA branch. Both of them are the same as those in the reviewer's Equation (2). B_T is the fitting coefficient of the new term. This new term enables Equation (3-1) to capture the main characteristic of the TA branches in simulation, as shown in Fig. R3-1(b). It needs to be pointed out that the dispersion curves in Fig. R3-1(b) do not show the flattening of the phonon branches upon approaching the Brillouin zone boundary in simulation. This is a drawback, but it does not influence the boson peak, since the boson peak anomaly locates at the low-frequency range.

We then calculate the reduced VDOS according to the dispersion relations from the simulation and those from the theoretical model (only the contribution of TA branch is considered here), respectively. Impressively, as shown in Figs. R3-1(c) and R3-1(d), the reduced VDOS by simulation and the theoretical model exhibits the same change tendency: the frequency of the boson peak increases with the weakening of phonon softening. It means that the theoretical model mentioned by the reviewer well captures the fundamental physics of the BP-like anomaly in this work. As a result, there is a nice marriage between our simulation and

this theory. The theoretical model further lends credence to our key point that the BP-like anomaly is caused by the phonon softening in strain glass.

Figure R3-1. Comparison of phonon dynamics obtained by simulation and the theoretical framework without assumptions of disorder. **a.** [001] TA branches for different strain glass compositions (x) by simulation. **b.** TA branches calculated according to the equation (3-1). We set $\nu_T = 0.5$, $D_T = 0.012$ and dial A_T in the range of $[0.004, 0.007]$. B_T changes with A_T in the range of $[0.000055, 0.000025]$. **c.** Reduced VDOS $g(\omega)/\omega^2$ vs. ω for different x by simulation. **d.** Reduced VDOS $g(\omega)/\omega^2$ vs. ω according to the theoretical framework. We set $k_D = 11$. The contribution of longitudinal acoustic branch is omitted here.

Changes made in the revised manuscript:

1. Page 28, Line 625:
We added Fig. R3-1 into the revised manuscript as Fig. 6.
2. Page 11, Line 294:
We added Equation (3-1) into the revised manuscript and gave a detailed description of the theoretical model in the discussion part.

Comment 5:

4. *Following on this direction, I think the authors should be more precise about the importance and the main messages of their results. (I) The theoretical models relying on disorder cannot explain the BP in this system, (II) the idea of the BP being the VH is clearly falsified by these experimental results (as written by the authors in line 152), (III) the results clearly support the theoretical model of [2, 1] as a viable explanation for the BP in this system. These three points must be clearly stated in the abstract, since are the major implications of the authors analysis.*

Response:

We thank the reviewer for helping us to summarize the main messages of our results. Indeed, these main messages were not clearly stated in the previous version. According to this comment, we made a substantial revision to the abstract to include all these three key points.

Changes made in the revised manuscript:

1. Page 1, Line 21:

We added, “This anomaly neither is a relic of van Hove singularity nor can be explained by other theories relying on disorder, while it verifies a recent theoretical model without any assumptions of disorder.”

Comment 6:

5. *In the context of models which predict a BP from phonon softening (i.e. softening of shear modulus) in both glasses and perfectly ordered non-centrosymmetric crystals, and crystals with locally non-centrosymmetric nearest-neighbour force imbalance, also Ref. [4] should be mentioned and discussed. This reference also showed that the van Hove singularity is distinct from the BP in crystals.*

Response:

We thank the reviewer for this comment. The Ref. [4] has now been cited and properly acknowledged. Moreover, other references mentioned by the reviewer have also been cited.

Comment 7:

Final evaluation

In my opinion, the manuscript represents a solid piece of work with a potential impact both from the experimental but also from the theoretical/conceptual side, provided the authors make an effort to offer a more quantitative interpretation of their results, as explained above. The work constitutes potentially strong evidence towards a complete re-thinking process about glass-like anomalies in solids, and it is an experimental proof that almost all the existing theories for BP anomalies fail in capturing its origin. It moreover appears in nice agreement with recent theoretical progress opening a new path towards the understanding of glassy anomalies in ordered phases.

I am happy to reconsider this manuscript for publication in Nature Communications after important revisions regarding the points above are made.

References

- [1] M. Baggioli and A. Zaccone. Unified theory of vibrational spectra in hard amorphous materials. *Phys. Rev. Research*, 2:013267, Mar 2020.
- [2] Matteo Baggioli and Alessio Zaccone. Universal origin of boson peak vibrational anomalies in ordered crystals and in amorphous materials. *Phys. Rev. Lett.*, 122:145501, Apr 2019.
- [3] J. F. Gebbia, M. A. Ramos, D. Szewczyk, A. Jezowski, A. I. Krivchikov, Y. V. Horbatenko, T. Guidi, F. J. Bermejo, and J. Ll. Tamarit. Glassy anomalies in the low-temperature thermal properties of a minimally disordered crystalline solid. *Phys. Rev. Lett.*, 119:215506, Nov 2017.
- [4] R. Milkus and A. Zaccone. Local inversion-symmetry breaking controls the boson peak in glasses and crystals. *Phys. Rev. B*, 93:094204, Mar 2016.
- [5] Manuel Moratalla, Jonathan F. Gebbia, Miguel Angel Ramos, Luis Carlos Pardo, Sanghamitra Mukhopadhyay, Svemir Rudi_c, Felix Fernandez-Alonso, Francisco Javier Bermejo, and Josep

Lluís Tamarit. Emergence of glassy features in halomethane crystals. *Phys. Rev. B*, 99:024301, Jan 2019.

[6] G. Remenyi, S. Sahling, K. Biljaković, D. Starešinić, J.-C. Lasjaunias, J. E. Lorenzo, P. Monceau, and A. Cano. Incommensurate systems as model compounds for disorder revealing low-temperature glasslike behavior. *Phys. Rev. Lett.*, 114:195502, May 2015.

[7] M. A. Strzemechny, A. I. Krivchikov, and A. Jeowski. Heat capacity of molecular solids: The special case of cryocrystals. *Low Temperature Physics*, 45(12):1290-1295, 2019.

Response:

We deeply thank the reviewer for commenting this work to be *a solid piece of work with a potential impact both from the experimental but also from the theoretical/conceptual side*. Moreover, we also totally agree with the reviewer that it requires a complete re-thinking process about glasslike anomalies in solids, since these glasslike anomalies, like the boson peak, are even more universal beyond the range of structural glasses. We are glad that this work exhibits a nice match with the theoretical framework pointed out by the reviewer, which lends further support to our key point. After a major revision according to the reviewer's very helpful suggestions, we hope the new manuscript may help to develop a deeper understanding of the origin of glassy anomalies in solids both with disorder and without.

Summary

In summary, we express our deep appreciation to all the three reviewers again for their careful reviews of this manuscript and their various constructive comments and suggestions. We have incorporated their suggestions into the revised manuscript as appropriate, and we hope the improved paper is now acceptable for publication in Nature Communications.

REVIEWERS' COMMENTS

Reviewer #1 (Remarks to the Author):

The authors have fully addressed my previous comments and significantly improved the manuscript. The calculations of the spectral functions and their agreement with previous inelastic neutron scattering results are particularly convincing. I now fully support publication of the manuscript.

Reviewer #2 (Remarks to the Author):

Still highly appreciating the results of this study, I am still not in a position to accept discussion of the results in terms of Boson peak.

The Boson peak in glasses is an anomaly, because the peak in density of phonon states, at least for the first glance, does not correspond to any flat part of dispersion relations. In this study, on the contrary, the peak is perfectly explained by dispersion relations. Thus, the peak is NOT an anomaly (contrary to what is stated in the manuscript), but one of many other low-energy peaks, perfectly corresponding to dispersion relations.

A simple observation of a low-energy peak in a crystal is not sufficient to suggest the relevant effect as a possible explanation of the Boson peak in glasses. For example, in clathrates, a low-energy peak in DOS appears because of guest atom rattling in a host matrix, and there is no sense to suggest that rattling mode is a common feature of vibrations in glasses. Similarly, there is no indication that the effect observed in this study can be relevant to all glasses. At least, authors do not provide any arguments for such generalization.

Thus, instead of providing new insight of dynamics of glasses, the manuscript rather looks like providing new confusion. Therefore, highly appreciating the results of this study, I am not in a position to recommend the manuscript for publishing.

Reviewer #3 (Remarks to the Author):

I would like to thank the Authors for the effort put in the revision of the manuscript which is undeniable. I have carefully read the revised manuscript and the reply to my observations and those of the other Referees as well.

I believe the authors did a great job in replying to all my concerns and in improving their manuscript. In particular, in addition to the nice experimental results, the paper now contains a clear and strong physical interpretation (that of phonons softening vs the standard Ioffe-Regel picture due to structural disorder) which goes definitely beyond the standard paradigm of the BP in glasses and it can be extremely relevant to explain other recent experimental observations. The conclusions are now better supported by the improved analysis and they could definitely open a new direction, or at least a fresh view, on an unsolved old problem as the BP.

In summary, I suggest the publication of this manuscript in Nature Communications.

Responses to Reviewers' Reports on NCOMMS-20-40100A

Response to Reviewer 1

Comment: *The authors have fully addressed my previous comments and significantly improved the manuscript. The calculations of the spectral functions and their agreement with previous inelastic neutron scattering results are particularly convincing. I now fully support publication of the manuscript.*

Response:

We thank the reviewer for his/her support for the publication of this manuscript. We particularly appreciate the crucial suggestions of the reviewer, which are very helpful and make this manuscript much more convincing.

Response to Reviewer 2

Comment: *Still highly appreciating the results of this study, I am still not in a position to accept discussion of the results in terms of Boson peak.*

The Boson peak in glasses is an anomaly, because the peak in density of phonon states, at least for the first glance, does not correspond to any flat part of dispersion relations. In this study, on the contrary, the peak is perfectly explained by dispersion relations. Thus, the peak is NOT an anomaly (contrary to what is stated in the manuscript), but one of many other low-energy peaks, perfectly corresponding to dispersion relations.

A simple observation of a low-energy peak in a crystal is not sufficient to suggest the relevant effect as a possible explanation of the Boson peak in glasses. For example, in clathrates, a low-energy peak in DOS appears because of guest atom rattling in a host matrix, and there is no sense to suggest that rattling mode is a common feature of vibrations in glasses. Similarly, there is no indication that the effect observed in this study can be relevant to all glasses. At least, authors do not provide any arguments for such generalization.

Thus, instead of providing new insight of dynamics of glasses, the manuscript rather looks like

providing new confusion. Therefore, highly appreciating the results of this study, I am not in a position to recommend the manuscript for publishing.

Response:

We thank the reviewer for her/his highly appreciating the results of this study. We understood the concerns and confusion of the reviewer about the interpretation of the BP-like anomaly in strain glass. According to the editor's suggestions, we have added a paragraph at the end of the discussion part of the revised manuscript, and try to solve this confusion by explaining how the BP anomaly is defined for strain glass and structural glasses and what is the key differences in the BP anomaly between strain glass and structural glasses.

In structural glasses, the boson peak is generally defined as an excess of vibrational states in the VDOS, departing from either the Debye squared-frequency law or their long-range ordered counterparts (i.e., crystals) [Nature 422, 289 (2003); Phys. Rev. Lett. 122, 145501 (2019)]. It can be manifested by either the low-frequency peak in a plot of the reduced VDOS $g(\omega)/\omega^2$ vs. ω or the low-temperature hump in a plot of C_p/T^3 vs. T . Similarly, the BP-like anomaly in this work is defined as the excess of vibrational states in the VDOS beyond its corresponding long-range ordered phase (i.e., martensitic phase). As shown in the manuscript, the anomaly found in strain glass matches these characteristic features of the boson peak in structural glasses in both VDOS and C_p . As a result, from the view of the phenomenon, the anomalies in strain glass and structural glasses are similar.

The key differences lie at the underlying mechanism. In structural glasses, phonon damping plays a fundamental role in giving rise to the boson peak, while the boson peak appears at a low frequency region where the dispersion relation is still linear. In comparison, the BP-like anomaly in strain glass stems from a nonlinear TA branch with phonon softening, while the influence of phonon damping is weak. We can get the fundamental difference in a more explicit way by referring to the theoretical model pointed out by Reviewer 3 in the last round [Phys. Rev. Lett. 122, 145501 (2019); Phys. Rev. Research 2, 013267 (2020)], namely the Green's function of Equation (2) in the manuscript, which is

$$G_\lambda(k, \omega) = \frac{1}{\omega^2 - \Omega_\lambda^{*2}(k) + i\omega\Gamma_\lambda^*(k)},$$

where λ represents either the longitudinal $\lambda=L$ or transverse $\lambda=T$ displacement field. $\Omega_{\lambda}^*(k)$ is the propagating term, representing the dispersion relation of acoustic phonons. $\Gamma_{\lambda}^*(k)$ is the damping coefficient, representing the diffusive phonon damping for acoustic phonons.

According to the equation above, the boson peak in structural glasses is mainly determined by a strong damping coefficient $\Gamma_{\tau}^*(k)$, while the propagating term $\Omega_{\tau}^*(k)$ is linear. In comparison, the BP-like anomaly in strain glass is mainly governed by the nonlinear propagating term $\Omega_{\tau}^*(k)$, while the damping coefficient $\Gamma_{\tau}^*(k)$ is very small.

Therefore, in this work we only suggest that BP-like anomaly can be triggered by a new mechanism of phonon softening, and we do not deny the fundamental role of phonon damping in understanding the origin of BP in structural glasses. Our work actually provides a fresh view to understand the origin of the boson peak, as pointed out by Reviewer 3.

To sum up, we acknowledge the concerns of reviewer 2, and provide a clear definition of the BP-like anomaly in strain glass, and present a much more explicit discussion about the fundamental differences between the BP anomaly of strain glass and structural glasses. Meanwhile, we also make a clear statement that the mechanism proposed in this work is not necessarily apply to all glasses in general.

Change made in the revised manuscript:

1. Page 13, Line 345:

We added a paragraph to solve the reviewer's concerns:

“We note that the interpretation of the BP-like anomaly in strain glass does not necessarily apply to all glasses in general. The boson peak in structural glasses is defined as an excess of vibrational states in the VDOS as compared with either the Debye squared-frequency law or their long-range ordered counterpart (i.e., crystals)²⁰. Similarly, the BP-like anomaly of strain glass in this work is defined as the excess states beyond its corresponding long-range ordered phase (i.e., martensitic phase). However, the underlying mechanisms of the two anomalies are different. For structural glasses, the boson peak is tightly related to phonon damping and takes place in a frequency region where the dispersion relation is still linear. In contrast, the BP-like anomaly in strain glass is tied to a nonlinear TA

branch exhibiting phonon softening. The key differences can be presented in a more explicit way according to the Green's function of Equation (2): the boson peak in structural glasses is mainly governed by a strong damping coefficient $\Gamma_{\text{T}}^*(k)^{29, 30}$, whereas the BP-like anomaly in strain glass is mostly dominated by a nonlinear propagating term $\Omega_{\text{T}}^*(k)$. Our work does reveal that the BP-like anomaly can be triggered by a different mechanism but does not deny the fundamental role of phonon damping in understanding the origin of the boson peak in structural glasses.”

Response to Reviewer 3

Comment: *I would like to thank the Authors for the effort put in the revision of the manuscript which is undeniable. I have carefully read the revised manuscript and the reply to my observations and those of the other Referees as well.*

I believe the authors did a great job in replying to all my concerns and in improving their manuscript. In particular, in addition to the nice experimental results, the paper now contains a clear and strong physical interpretation (that of phonons softening vs the standard Ioffe-Regel picture due to structural disorder) which goes definitely beyond the standard paradigm of the BP in glasses and it can be extremely relevant to explain other recent experimental observations. The conclusions are now better supported by the improved analysis and they could definitely open a new direction, or at least a fresh view, on an unsolved old problem as the BP.

In summary, I suggest the publication of this manuscript in Nature Communications.

Response:

We thank the reviewer for the recommendation for the publication. The current manuscript benefits a lot from the constructive comments and insightful suggestions of the reviewer, and we really appreciate his/her effort in helping us improve the quality of this work substantially.